# TS1 Global evaluation of the "dry gets drier, and wet gets wetter" paradigm from a terrestrial water storage change perspective

**Jinghua Xiong**[1]**, Shenglian Guo**[1]**, Abhishek**[2]**, Jie Chen**[1]**, and Jiabo Yin**[1]

[1]State Key Laboratory of Water Resources and Hydropower Engineering Science, Wuhan University, Wuhan, 430072, China
[2]School of Environment and Society, Tokyo Institute of Technology, Yokohama 226-8503, Japan

**Correspondence:** Shenglian Guo (slguo@whu.edu.cn)

**Abstract.** The "dry gets drier, and wet gets wetter" (DDWW) paradigm has been widely used to summarize the expected trends of the global hydrologic cycle under climate change. However, the paradigm is largely conditioned by choice of different metrics and datasets used and is still comprehensively unexplored from the perspective of terrestrial water storage anomalies (TWSAs). Considering the essential role of TWSAs in wetting and drying of the land system, here we built upon a large ensemble of TWSA datasets, including satellite-based products, global hydrological models, land surface models, and global climate models to evaluate the DDWW hypothesis during the historical (1985–2014) and future (2071–2100) periods under various scenarios with a 0.05 significance level (for trend estimates). We find that 11.01 %–40.84 % (range by various datasets) of global land confirms the DDWW paradigm, while 10.21 %–35.43 % of the area shows the opposite pattern during the historical period. In the future, the DDWW paradigm is still challenged, with the percentage supporting the pattern lower than 18 % and both the DDWW-validated and DDWW-opposed proportion increasing along with the intensification of emission scenarios. We show that the different choices of data sources can reasonably influence the test results up to a 4-fold difference. Our findings will provide insights and implications for global wetting and drying trends from the perspective of TWSA under climate change.

## 1 Introduction

The global hydrological cycle has experienced considerable changes due to climate change and anthropogenic interventions, exerting a tremendous impact on agriculture, ecological environment, and freshwater availability globally (Shugar et al., 2020; Perera et al., 2020; Gampe et al., 2021). Assessing the variations of constituent components of the water cycle, namely, precipitation ($P$), evapotranspiration ($E$), runoff ($R$), and storage change, is therefore crucial in understanding the systematic hydrological response and dealing with water-related issues in the context of global change (Moreno-Jimenez et al., 2019; Zhao et al., 2021; Yin et al., 2022). Under these circumstances, the "dry gets drier, and wet gets wetter" (DDWW) paradigm, firstly introduced by Held and Soden (2006), has become one of the most widely used hypotheses to summarize the long-term trends in the global hydrological cycle (Roderick et al., 2014; Yang et al., 2019). Initially, it was developed based on the deficit between precipitation and evapotranspiration ($P - E$), which is expected to increase due to the enhancement of atmospheric water vapor in humid regions (i.e., convergence zones) under a warming climate and decrease over arid regions (i.e., divergence zones) (Durack et al., 2012). The DDWW paradigm has been used to represent the historical and future trends in various constituent components of the hydrologic cycle on regional (Chou et al., 2009; Allan et al., 2010; Hu et al., 2019; Zeng et al., 2019) and global scales (Held and Soden, 2006; Donat et al., 2016). However, the rationale and validity of the DDWW mechanism are recently questioned at different levels through the growing number of datasets, model simulations, and indicators (Polson and Hegerl, 2017; Yang et al.,

2019; Y. Li et al., 2021). Byrne and O'Gorman (2015) used simulations from 10 climate models to reveal an ocean–land contrast pattern in the response of $P - E$ to global warming in historical (1976–2005) and future (2071–2099) periods, highlighting the DDWW as a more suitable mechanism over ocean than over land. Given the fact that historical evaluation of the DDWW paradigm was mainly based on oceanic observations, Greve et al. (2014) adopted 2142 possible combinations of $P - E$ to assess the trends in wetting and drying over global land and discovered merely 10.8 % of the area following the DDWW pattern during the 1948–2005 period. Roderick et al. (2014) revisited the DDWW paradigm, cautioned about its interpretation owing to the different behavior of land and ocean with respect to the water cycle, and showed that the paradigm does not hold true in terms of projected changes in the mean annual water balance over land. Alternatively, Yang et al. (2019) integrated an ensemble of six hydroclimatic indicators for the global assessment of the DDWW paradigm between 1982 and 2012, suggesting the phenomenon only occurred over 20 % of the global land. In a nutshell, there are great uncertainties still remaining in the assessments and subsequent interpretation of global trends in dryness and wetness under climate change (Dai, 2011; Trenberth et al., 2014).

The uncertainties within previous studies are mainly sourced from different choices of metrics adopted and datasets used for evaluating the changes in dryness and wetness (Vicente-Serrano et al., 2010; Feng and Zhang, 2015; Huang et al., 2016). Specifically, the widely used metric $P - E$ over the ocean has been proven overwhelmingly positive over land based on both observations and simulations, revealing an ocean-dominated behavior (Greve et al., 2014; Byrne and O'Gorman, 2015; Greve and Seneviratne, 2015). Moreover, some meteorological indices derived from precipitation and evapotranspiration, such as the standardized precipitation evapotranspiration index (SPEI), aridity index (AI), and standardized precipitation/evapotranspiration index (SPI/SETI), do not capture the integrated response of the land system due to the trade-off between the simplicity of meteorological factors and computational requirements of process-based variables (Huntington, 2006; Dai, 2011; Slette et al., 2020; Barnard et al., 2021). A few indices like the standardized soil moisture index (SSI), standardized groundwater index (SGI), and standardized runoff index (SRI), however, focus on a single aspect of the water cycle and do not describe the integrated status of the terrestrial water storage (TWS) (AghaKouchak, 2014; Wu et al., 2018; Guo et al., 2021). In coupled human–natural systems, where the synergistic impacts of natural and anthropogenic drivers are exceedingly difficult to disentangle, an integrated representation of the land systems is of paramount importance for policymakers (Rodell et al., 2018).

TWS, consisting of water storage in surface water, soil moisture, groundwater, snow and ice, and canopies, can physically provide integrated information about the overall status of the land, whose changes are closely linked to the terrestrial wetting and drying tendency (Tapley et al., 2019; Pokhrel et al., 2021). Apart from the societal and economic importance, TWS plays a vital role in Earth system processes, including climate, weather, and biogeochemical cycles (Abhishek et al., 2021). Change in storage, i.e., the difference between the consecutive TWS values, is a key variable of the hydrological cycle. Therefore, understanding the spatiotemporal dynamics of past and future TWS is not only essential for human life, but also crucial for assessing the water cycle, planning, policymaking, and other management strategies for water resources in a changing climate and for a continuously increasing population (Abhishek et al., 2021). There are several studies dealing with TWS or derived indicators to assess freshwater availability (Rodell et al., 2018), water storage dynamics (Scanlon et al., 2018), and droughts and flood monitoring (Abhishek et al., 2021; Long et al., 2014), among others. Divergent patterns of TWS changes have been reported over arid and humid regions under the combined effects of climate change (e.g., global warming), climatic variability (e.g., ENSO), and human activity (e.g., groundwater pumping) (Chang et al., 2020; An et al., 2021; Hu et al., 2021). However, there is no study to comprehensively examine the global variability and validity of the DDWW paradigm in the past and future in terms of TWS changes. Furthermore, divergent datasets produce different trends in TWS due to distinctive internal variability and external forcing (from satellites and meteorological stations), especially from precipitation and evapotranspiration (Chen et al., 2020). For example, Scanlon et al. (2018) conducted comprehensive comparisons between decadal trends in TWS from seven global models and three Gravity Recovery and Climate Experiment (GRACE) satellite solutions over major basins globally and showed a large underestimation of the increasing and decreasing trends of models primarily due to human water use and forcing climate variations.

Therefore, to bridge the aforesaid research gap, we conduct a systematic evaluation of the DDWW paradigm from the perspective of terrestrial water storage anomalies (TWSAs) using an ensemble of five different TWS datasets, including one GRACE reconstruction, two global hydrological models (GHMs), and two land surface models (LSMs) between 1985 and 2014. Subsequently, an alternative ensemble of eight global climate models (GCMs) from the Coupled Model Intercomparison Project 6 (CMIP6) is used to further test the paradigm under various scenarios during the future period (2071–2100). Utilizing the data from these models and observation-based products, we further establish the metric "$P - E - R$" in terms of the water balance equation for intercomparisons with the test results from the aspect of the TWSA and for highlighting the governing mechanisms of the estimated disparities.

## 2 Data and methods

### 2.1 Data preprocessing

We perform the assessment of the DDWW paradigm over global land at both gridded $1° \times 1°$ cell and regional scales, excluding Greenland and Antarctica. One of the global hotspots with significant changes in hydroclimatological conditions (e.g., precipitation and air temperature) (Liu et al., 2006; Zhang et al., 2017), i.e., the Qinghai–Tibetan Plateau (QTP), is selected as a typical region for regional analysis because it has experienced alarming TWS losses in recent decades and shows continuing declines under future scenarios (Meng et al., 2019; Li et al., 2022). The QTP and its surroundings which are called the world's "Third Pole" play a crucial role in the freshwater availability of more than 1.4 billion people (Immerzeel et al., 2010). The QTP is mainly covered by polar tundra and is a cold and arid steppe climate region (Fig. S2 in the Supplement), causing the sparse distribution of in situ networks there (Wan et al., 2014). Thus, using alternative methods such as remote sensing (e.g., GRACE) and global model outputs (e.g., GHMs, LSMs, and GCMs) to study the hydrological variations in the QTP is of much importance.

We use an ensemble of five TWSA datasets to evaluate the DDWW paradigm during the historical period 1985–2014, which includes one GRACE reconstruction, two global hydrological models (GHMs), and two global land surface models (LSMs) (see Table 1 and next sections). Please note that some studies may use the term GHMs to represent both global hydrological and water resource models (GH-WRMs) and LSMs together (Scanlon et al., 2018), while we use it only for the former one for distinction and simplicity. Since no dataset presents the absolutely "true" value, we demonstrate the individual results of each member to avoid the uncertainty derived from different TWSA definitions in various models/products (Supplement Table S1). The missing months (12 % of the months, i.e., June 2002, July 2002, June 2003, January 2011, June 2011, May 2012, October 2012, March 2013, August 2013, September 2013, February 2014, July 2014, and December 2014) of GRACE measurements have been filled using a linear interpolation method. In addition, an ensemble of eight TWSA simulations from CMIP6 GCMs is used to examine the DDWW paradigm in the future period (2071–2100). The members of the CMIP6 ensemble and all of the historical datasets have been resampled to a $1° \times 1°$ scale using a bilinear interpolation approach for consistency and better comparison in the spatial domain. The ensemble mean of CMIP6 models has been estimated using simple averaging because they have the same simulation objects (Table S1). All the historical datasets and CMIP6 members, as well as their ensemble, are represented as the long-term anomaly relative to the baseline between 1985 and 2014. We also calculate the metric $P - E - R$ based on the water balance equation for cross-comparison with the test results from the TWSA perspective. This metric is estimated using $P$, ET, and $R$ from the same models as those of TWSA (e.g., GHMs, LSMs, and GCMs) for consistency. Moreover, an observation-based combination is also derived as a benchmarking subset based on precipitation ($P$) from the Climatic Research Unit gridded Time Series (CRU TS-v4.06; Harris et al., 2020), evapotranspiration ($E$) from the Global Land Evaporation Amsterdam Model (GLEAM-v3.6; Martens et al., 2017), and runoff ($R$) from the G-RUN ENSEMBLE (Ghigg et al., 2021) (Table 1).

### 2.1.1 GRACE and GRACE reconstructions

The GRACE (and GRACE Follow-On) missions have provided unprecedented estimates of monthly TWSAs worldwide from April 2002 up to the present though with the 33 months missing because of instrumental issues and mission interruption (Tapley et al., 2004). We use the GRACE mascon solution from the Center for Space Research at the University of Texas at Austin (UTCSR) to serve as the benchmarking product from the period 2002–2014 (Watkins et al., 2015). Compared to conventional GRACE products (e.g., spherical harmonic solutions), mascon solutions do not need spatial (e.g., smoothing) or spectral (e.g., de-striping) filtering or other empirical scaling and therefore have a higher signal-to-noise ratio, higher spatial resolutions, and eventually reduced errors (Save et al., 2016; Watkins et al., 2015). However, the GRACE observational products were not adequate to assess the long-term trends of TWSAs due to relatively short temporal coverage ($\sim 20$ years). Therefore, we obtain the GRACE reconstruction provided by F. Li et al. (2021) for evaluation of the DDWW paradigm, which is generated using state-of-the-art machine learning and statistical methods and is also trained by the consistent GRACE mascon product from the UTCSR institution. The GRACE reconstruction applies four meteorological variables (i.e., precipitation, 2 m air temperature, sea surface temperature, and multiple climate indices) and three hydrological variables (i.e., soil moisture, runoff, and evaporation) to simulate the temporally decomposed GRACE signals (i.e., the seasonal, interannual, and residual components) (F. Li et al., 2021). We would like to mention that the linear trend components in GRACE reconstructions are directly added by the linear GRACE trends, which are mainly caused by glacier melt and anthropogenic factors (e.g., dam constructions and water abstractions). These factors are difficult to predict using the climatic and hydrologic inputs and may change over time (e.g., interannual and decadal variability), causing the possible bias in the long-term trend estimates from GRACE reconstructions. The accuracy and applicability of the GRACE reconstruction have been fully evaluated over global land in several previous studies (Xu et al., 2021; Yi et al., 2021).

**Table 1.** Datasets used in this study.

| Type | Data | URL | Selected period | Raw temporal resolution | Raw spatial resolution (latitude × longitude) |
|---|---|---|---|---|---|
| GRACE reconstructions | F. Li et al. (2021) | https://doi.org/10.1029/2021GL093492 | 1985–2014 | Monthly | 0.5° × 0.5° |
| GRACE observations | GRACE CSR RL06 mascons-v02 | http://www2.csr.utexas.edu/grace/ | 2002–2014 | Monthly | 0.25° × 0.25° |
| GHMs | WGHM-v2.2d | https://doi.org/10.5194/gmd-14-1037-2021 | 1985–2014 | Monthly | 1° × 1° |
| | GLDAS2.0-VIC | https://ldas.gsfc.nasa.gov/gldas | 1985–2014 | Monthly | 1° × 1° |
| LSMs | GLDAS2.0-Noah | https://ldas.gsfc.nasa.gov/gldas | 1985–2014 | Monthly | 1° × 1° |
| | GLDAS2.0-CLSM | https://ldas.gsfc.nasa.gov/gldas | 1985–2014 | Monthly | 1° × 1° |
| GCMs | ACCESS-CM2 | https://esgf-node.llnl.gov/projects/cmip6/ | 1985–2100 | Daily/monthly | 1.25° × 1.875° |
| | ACCESS-ESM1-5 | https://esgf-node.llnl.gov/projects/cmip6/ | 1985–2100 | Daily/monthly | 1.24° × 1.875° |
| | CanESM-5 | https://esgf-node.llnl.gov/projects/cmip6/ | 1985–2100 | Daily/monthly | 2.8125° × 2.8125° |
| | GFDL-ESM4 | https://esgf-node.llnl.gov/projects/cmip6/ | 1985–2100 | Daily/monthly | 1° × 1.25° |
| | IPSL-CM6A-LR | https://esgf-node.llnl.gov/projects/cmip6/ | 1985–2100 | Daily/monthly | 1.2587° × 2.5° |
| | MIROC6 | https://esgf-node.llnl.gov/projects/cmip6/ | 1985–2100 | Daily/monthly | 1.4063° × 1.4063° |
| | MPI-ESM1-2-HR | https://esgf-node.llnl.gov/projects/cmip6/ | 1985–2100 | Daily/monthly | 0.9375° × 0.9375° |
| | MPI-ESM1-2-LR | https://esgf-node.llnl.gov/projects/cmip6/ | 1985–2100 | Daily/monthly | 1.875° × 1.875° |
| Observation-based precipitation and potential evapotranspiration | CRU TS-v4.06 | https://crudata.uea.ac.uk/cru/data/hrg/cru_ts_4.06/ | 1985–2014 | Monthly | 0.5° × 0.5° |
| Observation-based runoff | G-RUN Ensemble | https://doi.org/10.1029/2020WR028787 | 1985–2014 | Monthly | 0.5° × 0.5° |
| Satellite-based evapotranspiration | GLEAM-v3.6a | https://www.gleam.eu/ | 1985–2014 | Monthly | 0.25° × 0.25° |

All links in the table were last accessed on 2 December 2022.

### 2.1.2 Global hydrological models

We use two global hydrological models, including the Variable Infiltration Capacity macroscale model (VIC-v4.1.2) and the WaterGAP hydrological model (WGHM-v2.2d),
to estimate TWS and $P - E - R$ for independent evaluation of the DDWW paradigm. The physically based, semi-distributed, and grid-based VIC model is managed by the NASA Global Land Data Assimilation System Version 2.0 (GLDAS-v2.0) (Liang et al., 1994; Syed et al., 2008). Forced
by the Global Data Assimilation System atmospheric analysis fields (Derber et al., 1991) and the Air Force Weather Agency's AGRicultural METeorological modeling system radiation fields, the VIC model can effectively capture the terrestrial water cycle by simulating the water stored in the
canopies, snow, and soil moisture within three soil layers up to a depth of 200 cm. The VIC model has been widely used to analyze terrestrial water storage changes at regional and global scales (Hao and Singh, 2015; Hao et al., 2018). The WGHM is a grid-based global hydrological model quanti-
fying the human water use and continental water fluxes for all land areas excluding Antarctica (Müller Schmied et al., 2021). Unlike most global hydrological models, the WGHM forced by the ERA40 and ERA-Interim reanalysis can simulate groundwater storage by coupling with global water
use models and linking model Groundwater-Surface Water Use (GWSWUSE), suggesting a comparably better representation of TWS (Döll et al., 2014). Several frequently used model outputs such as TWS, discharge, and water use have been evaluated against global observations (Wan et al.,
2021). $E$ and $R$ from the VIC and WGHM models are also extracted for the calculation of the variable "$P - \mathrm{ET} - R$" by combining the $P$ from their meteorological inputs of GLDAS2.0.

### 2.1.3 Land surface models

We use two land surface models consisting of the Noah (v3.6) and Catchment (CLSM-vF2.5) models to calculate TWS and $P - E - R$ globally for parallel assessment of the DDWW paradigm. Similar to the VIC model, both Noah and CLSM models are managed by GLDAS (v-2.0) from
the NASA GSFC institute. GLDAS is a composite of global hydrological and land surface models that simulate the optimal fields of the land by using state-of-the-art data assimilation and land surface simulation techniques (Rodell et al., 2004). GLDAS has been widely used to compare with
GRACE TWSA in data-sparse regions such as Africa and the Qinghai–Tibetan Plateau (Ogou et al., 2022; Xing et al., 2021). The Noah-modeled TWS is considered the sum of canopy water storage, snow water equivalent, and soil moisture of four layers with a total depth of 200 cm. Different
from that, the CLSM simulates shallow groundwater, and the vertical levels of soil moisture are not explicitly divided within the depth of 100 cm. Similarly, we used the $E$ and $R$

modeled by the CLSM and Noah models to calculate the index $P - E - R$. We note that the three GLDAS models (i.e., VIC, CLSM, and Noah) share the same $P$ estimations due to
55 the consistent meteorological inputs, which might reduce the bias in the estimates of the metric $P - E - R$.

### 2.1.4 Global climate models

We use a suite of eight global climate models belonging to the ensemble "r1i1p1f1" of CMIP6 to evaluate the DDWW
paradigm under climate change. The CMIP6 serves as a category of experiments of GCMs coupled to the dynamic ocean, simple land surface, and thermodynamic sea ice (Eyring et al., 2016). We choose these eight models out of the 34 CMIP6 models because they are the only models for which
TWSA outputs are available in both the historical and future periods under multiple emission scenarios (see Table 1). The CMIP6 (CMIP5) TWSA represents the sum of total soil moisture and snow water equivalent, which has been comprehensively validated with the GRACE data, though with em-
bedded uncertainties, over global major river basins (Freedman et al., 2014; Wu et al., 2021). The CMIP6 comparisons have become a diagnostic tool to better understand climate change in past, present, and future periods (Eyring et al., 2016), which includes a total of five Shared Socioeconomic
Pathways (SSPs) representing global economic and demographic changes under different greenhouse gas emissions. We select three SSP scenarios including SSP126, SSP245, and SSP585, representing the green roads, middle of the road, and the highway road, respectively (Iqbal et al., 2021).
Since the GCMs have different TWSA definitions from the "actual" TWSA observed by GRACE (Table S1), we employ a trend-preserving method to perform bias correction combined with historical GRACE data. The trend-preserving method initially developed by Hempel et al. (2013) modi-
fies the monthly means of the simulated data to match the observed data using a constant offset between simulations and observations and has been widely used in the Intersectoral Model Intercomparison Project (ISIMIP2b). The detailed procedure of the bias correction for CMIP6 TWSA
has been described in detail in a recent study (Xiong et al., 2022a). To show the difference before and after the bias correction, we select two typical regions (i.e., Amazon and Mekong River basins) with abundant surface and groundwater resources (Pham-Duc et al., 2019). Of the two selected
basins, the Mekong River basin experiences severe human interventions such as groundwater pumping, dam constructions, and urbanization, while the Amazon River basin is considered one of the largest natural river basins with low impacts of human activities (Xiong et al., 2022b). It is dis-
covered that the GCM simulations without bias correction show obvious underestimations over two regions with large uncertainty, which have, however, significantly reduced after bias correction along with a lower spread range (Fig. S13). The amplitudes of the GCM series are adjusted to nearly the

same as GRACE data, with the long-term trends unaffected. It is noteworthy that the trend-preserving method would not affect the long-term trends of the GCM TWSA and, therefore, not influence our current DDWW evaluation results. In ₅ addition to the TWSA, we also derive the predictions of $P$, $E$, and $R$ for the construction of the $P - E - R$ to compare with TWSAs similar to those from GHMs and LSMs.

## 2.2 Detection of wetting and drying

The TWSA, consisting of the water volume stored in the land ₁₀ surface and subsurface, is applied to define the "wetting" and "drying" conditions of the landmass in this study. The nondimensional TWS drought severity index (TWS-DSI) is established at both $1° \times 1°$ grid cell and regional and global scales, which is normalized by the regional hydroclimato- ₁₅ logical variability because a given magnitude of TWS deficit could indicate different dryness and wetness conditions in different climate regions. TWS-DSI has clear classification categories based on the US Drought Monitor (USDM) and is suitable for comparing the dryness and wetness status for ₂₀ different locations and periods (Table S2). It has been widely used in hydrology and climate fields due to its simple structure and effective ability to capture drying and wetting conditions (Pokhrel et al., 2021). The monthly TWS-DSI is calculated for all ensemble members and their mean from CMIP6 ₂₅ as follows (Zhao et al., 2017):

$$\mathrm{TWS-DSI}_{i,j} = \frac{\mathrm{TWS}_{i,j} - \mu_j}{\sigma_j}, \qquad (1)$$

where $\mathrm{TWS}_{i,j}$ is the TWS value in year $i$ and month $j$, and $\mu_j$ and $\sigma_j$ denote the mean and standard deviation of the annual TWS in month $j$, respectively. We convert the ₃₀ monthly TWS-DSI into annual means to calculate the long-term trends using the linear regression method. We examine the first-order autocorrelation of each TWSA dataset using the Durbin–Watson test (Durbin and Watson, 1950, 1951). We find a total of 20 % (GRACE reconstruction), ₃₅ 43 % (WGHM), 41 % (VIC), 23 % (CLSM), 29 % (Noah), and 20 % (GCM) of the grid cells not presenting autocorrelation during 1985–2014, respectively (Fig. S1). For the future period, the percentage is 25 %, 26 %, and 22 % under the SSP126, SSP245, and SSP585 scenarios, respectively. In this ₄₀ case, the significance of the long-term trends is evaluated using the modified Mann–Kendall trend test at a 5 % level to avoid autocorrelation (Hamed and Rao, 1998). The modified Mann–Kendall method uses the lag 1 autocorrelation coefficients to perform the bias correction for the data variance, in ₄₅ which only the significant lags (at a 0.05 level) are selected. However, the original Mann–Kendall method would be used if the selected lags cannot facilitate the variance correction well. Similarly, we also estimate the long-term trends of the index $P - E - R$ for comparison with TWS-DSI using the ₅₀ same methods. The area with a significant trend of increasing/decreasing TWS-DSI or $P - E - R$ is considered to be

undergoing wetting/drying; otherwise, it is defined as a region with a nonsignificant trend.

To evaluate the DDWW paradigm over global land, the effective aridity index (AI) is used to classify a grid cell ₅₅ as an arid, humid, and transitional region following Yang et al. (2019) because TWS-DSI/TWSA approximates zero for the long-term mean. The AI is calculated as the ratio of annual precipitation to potential evapotranspiration provided by the CRU TS-v4.06 during the same period as TWS-DSI ₆₀ (i.e., 1985–2014). The global distribution of multiyear average AI and the classifications during the 1985–2014 period is presented in Fig. S3, which is also highly consistent with the widely used Köppen–Geiger climate classification maps (Beck et al., 2018) (Fig. S2). It can be seen that most of the ₆₅ arid regions (AI < 0.5) are located in southwestern America, north and south Africa, central Asia, Arabian regions, and Australia, accounting for 39.3 % of the land. The percentage of humid areas (AI > 0.65) that are mainly located in eastern America, the Amazon region, central Africa, south- ₇₀ ern China, western Europe, and Russia reaches 52.8 % of the land. An approximate 7.9 % of the land area is defined as the transitional region, referring to an intermediate between arid and humid climates. The transitional region generally lies in the shared boundaries of the humid and arid re- ₇₅ gions (e.g., western America, northern Canada, central Asia, western Africa, eastern Russia, and Australia). The DDWW paradigm is evaluated at a 5 % significance level (trend estimates) in this study, combined with the standard AI-derived climate classifications. We calculate the global mean trends ₈₀ of TWS-DSI using a spatially weighted method to account for the changing area of grid cells with latitudes. The percentage of different change patterns (e.g., DD, dry gets drier, and WW, wet gets wetter) is calculated as the ratio of the corresponding land area to the global sum. Thus, a few miss- ₈₅ ing grid cells in datasets (6 %, 1 %, 3 %, and 1 % for GRACE reconstruction, WGHM, GLDAS, and GCMs, respectively) may marginally affect our final results.

## 3 Results and discussion

### 3.1 Global trends of dryness and wetness ₉₀

We firstly assess the reliability of the GRACE reconstruction, GHMs, and LSMs by comparing them with the GRACE observations. Figure S4 presents the global distribution of the normalized root mean square error (NRMSE) between the GRACE TWSA and different products during the pe- ₉₅ riod April 2002–December 2014, with the NRMSE calculated as the ratio of RMSE to the differences between the maximum and minimum GRACE TWSA. The GRACE reconstruction shows the best performance over five TWSA datasets, with the NRMSE generally lower than 0.2, with ₁₀₀ nearly half of the land area showing a NRMSE below 0.1. In particular, NRMSE ranging from 0.1 to 0.3 occurs in

western and central Asia, northern China, southern Australia, eastern Russia, northern and southern Africa, and central North and South America (Fig. S4). Two GHMs (i.e., WGHM and VIC) and two LSMs (CLSM and Noah) present a similar spatial pattern of NRMSE to the GRACE reconstruction but with a relatively higher bias, among which the VIC model outperforms the other three models. The CLSM model shows comparatively poor performance, which is also confirmed by the probability density distributions of NRMSE compared with GRACE (Fig. S4). The better performance of the GRACE reconstruction over other data may be because they are directly calibrated with the GRACE measurements during 2002–2017, while their performances need more validation beyond the GRACE era (i.e., prior to April 2002 and during July 2017–May 2018). A temporal comparison of global average TWSA derived from GHMs, LSMs, GRACE reconstruction, and CMIP6 and GRACE during 2002–2014 is shown in Fig. S5. The GRACE TWSA ranges from roughly −20 to 20 mm and shows obvious seasonal characteristics. A similar temporal pattern is captured by various models, with the change spread covering the variations of GRACE data. The NRMSE between multiple datasets and GRACE data ranges from 0.08 (GRACE reconstruction) and 0.16 (Noah), coinciding with the strong correlation within different datasets (Figs. S4 and S6). Moreover, the fluctuation range of the CMIP6 is generally larger than different historical models/products, highlighting the considerable uncertainty sourced from different forcing variables and model parameterizations. Then, we examine the difference between GCMs-simulated TWSA before and after the trend-preserving bias correction using GRACE. It is discovered that their correlation coefficients improve by comparing with GRACE, while slightly decreasing within the eight GCMs, which can be attributed to the introduced uncertainty when performing the bias correction (Fig. S7). In addition, the spatial distributions clearly show that the ensemble mean of eight GCMs outperforms each member globally, particularly in Australia, southern Africa, and North America (Figs. S8 and S9). The better performance becomes more obvious after bias correction. An overall decrease in NRMSE is also observed according to the probability density functions after performing bias correction, which is also detected from the Taylor diagram results (see Fig. S10). We also provide the evaluation of the bias-corrected TWSA changes (i.e., TWSC) using the water balance estimates (i.e., $P - E - R =$ TWSC) during 1985–2014 (Figs. S11 and S12). The observation-based water balance estimates correlate well with GRACE TWSA and GCM-modeled $P - E - R$ with a correlation coefficient of 0.62 and 0.93, respectively. The GCM-simulated changes in TWSA also present a strong correlation with the observed $P - E - R$ before and after bias correction. The spatial distribution of correlation coefficients between TWSC from observations and GCMs with and without bias correction shows that the performances in regions with good accuracy, like Alaska, western parts of the Tibetan Plateau, and

northern Russia, decrease after bias correction, which might be caused by the simplified treatment of permafrost in GCMs due to the prevailing uncertainties in, e.g., changes in thermophysical properties of the soil during freezing and thawing cycles (Burke et al., 2020). Conversely, the areas with relatively poorer accuracy before bias correction, such as northern Africa and northern South America, slightly improve after bias correction. Notwithstanding the observed differences in some regions, our trend-preserving method used for bias correction would not influence the long-term trend estimations of both TWSA and TWS-DSI and therefore does not impact our evaluation of the DDWW paradigm (Hempel et al., 2013). Although bias correction has been performed on the CMIP6 TWSA, some biases inherent to the uncertainty in parameters, hydrometeorological forcing, and internal variability of GCMs still exist, which may influence the assessment of the DDWW paradigm in the future period (2071–2100) climate change.

We assess the long-term trends of TWS-DSI during the historical period 1985–2014 (based on a GRACE reconstruction, two GHMs (WGHM and VIC), two LSMs (CLSM and Noah), and the ensemble mean of eight GCMs) and the future period 2071–2100 (based on the ensemble mean of eight GCMs) under SPSP126, SSP245, and SSP585 scenarios to provide insights into the terrestrial water storage changes for the DDWW paradigm (Figs. 1 and S14). The GRACE reconstruction, having the best accuracy among all other model-based TWSA, is selected for detailed analysis, which also shows the highest proportion of areas with significant trends. During the historical period, a clear spatial homogeneity (clustered patterns) of TWS-DSI trends is observed globally, and the average TWS-DSI has a significant decreasing slope of $-0.11 \, \text{yr}^{-1}$ ($p < 0.05$) (Fig. 1), similar to the results from SPI, SPEI, and AI (Wang et al., 2018; Yang et al., 2019), together with the results from other models (WGHM: $-0.07 \, \text{yr}^{-1}$, VIC: $-0.05 \, \text{yr}^{-1}$, CLSM: $-0.06 \, \text{yr}^{-1}$, Noah: $-0.04 \, \text{yr}^{-1}$, the ensemble mean of GCMs: $-0.05 \, \text{yr}^{-1}$). Spatially, severe drying ($p < 0.05$) exists on the coast of the Gulf of Alaska, the Canadian archipelago, Chile, and the QTP, with significant slopes of TWS-DSI ranging from $-0.09$ to $-0.12 \, \text{yr}^{-1}$ (Fig. 1), which is caused by the rapid melt of ice sheet, glacier ablation, and increase in the active permafrost layer under a warming climate (Luthcke et al., 2013; Velicogna et al., 2014). Triggered by severe historical droughts and extensive water use from groundwater and surface water over decades, the drying trends in northern Canada, southern California, and Texas can be clearly discovered, with a decreasing trend of TWS-DSI ranging from $-0.06$ to $-0.12 \, \text{yr}^{-1}$ ($p < 0.05$) (Bouchard et al., 2013; Haacker et al., 2016), as in eastern Brazil (Getirana, 2016). Moreover, overwhelming groundwater depletion due to unsustainable human water use such as irrigation is responsible for the increasing dryness at significant slopes, ranging from $-0.09$ to $-0.12 \, \text{yr}^{-1}$ in southeastern and northern regions of Africa, eastern and central Europe, central Asia, north-

ern China, and northern India (Rodell et al., 2009; Feng et al., 2013; Ramillien et al., 2014; Peña-Angulo et al., 2020; Xiong et al., 2022c). The decreasing TWS-DSI is also reported over European Russia because of the decline in the storage of surface and ground waters (Grigoriev and Frolova, 2018). Additionally, the significant decreases in TWS-DSI ranging from $-0.09$ to $-0.12\,\mathrm{yr}^{-1}$ ($p < 0.05$) around the Caspian and Aral seas are seen, which are from the reductions of inflow discharge and precipitation as well as evapotranspiration increase (Zmijewski and Becker, 2014). Naturally, a moderate drying trend in southwestern Africa and central Mediterranean Europe caused by precipitation decrease is detected by the reduction of TWS-DSI ($-0.06$ to $-0.12\,\mathrm{yr}^{-1}$) (Peña-Angulo et al., 2020). Conversely, increasing precipitation dominates the wetting trend in midlatitude regions, including southern Russia and Canada, western Africa, southeastern and southwestern Europe, southeast Asia, and northwestern China, with significant slopes roughly ranging from 0.06 to $0.12\,\mathrm{yr}^{-1}$ (Fig. 1) (Siebert et al., 2010; Ndehedehe et al., 2017; Peña-Angulo et al., 2020). Some regions, such as the Amazon River basin, south Africa and eastern Australia, presenting wetting trends, are considered to experience a climatic shift from the dry to the wet period (Chen et al., 2010; Gaughan and Waylen, 2012). When looking at the test results of the GHMs and LSMs, we notice the regional differences with generally consistent spatial patterns with the GRACE reconstruction. For example, the WGHM model shows depletion trends in TWS-DSI for the southwest of the South American continent. The three GLDAS models (i.e., VIC, CLSM, and Noah) do not capture the increasing trends in southern China (i.e., Yangtze and Pearl River basins), of which the VIC model surprisingly shows the increasing trends over the Arab region. We additionally compare the trend estimations of the GCMs' ensemble mean during the 1985–2014 period (Figs. 1 and S14). Despite the overall similarity to the above-mentioned datasets, the existing regional differences in western southern Africa (drying) and western Asia (wetting) compared with multiple models provide additional insights, indicating the great potential of the CMIP6 ensemble in TWSA projections.

Further, we perform an independent assessment based on the metric $P - E - R$ for comparison with the TWS-DSI results to reveal the inherent mechanisms of the changes (Figs. 2 and S15). The observational product of the variable $P - E - R$ presents a similar pattern to the test results using TWS-DSI though with nonsignificant trends over most regions. This can be explained by the fact that the magnitude of the changes in the water storage, i.e., TWSC, in a region is minimal compared to that of the TWSA trends (Lv et al., 2021). In particular, the decreasing $P - E - R$ (i.e., TWSC) in southwestern South America, northern and southern Africa, western Australia, northern China, European Russia, and central Asia is observed with trends $< -2\,\mathrm{mm\,yr}^{-1}$, while increasing trends in northern Canada, Central America, central Africa, eastern Australia, southern India, and south-

ern and eastern Russia are found with rates $> 2\,\mathrm{mm\,yr}^{-1}$. The local differences over the Arab region, south China, and the Caspian region might be caused by the propagated uncertainty in multiple observational datasets, especially for the arid regions (e.g., northern Africa and western America), where accurately estimating $E$ is very challenging (Goyal, 2004). For southern China, consisting of the Yangtze and Pearl River basins, the difference might arise from the extensive reservoir filling, such as the Three Gorges Dam (Zhong et al., 2009), highlighting the significant role of human activities in the regional variations of TWS. Similarities are also seen over the land around the Caspian Sea, which is largely affected by the direct diversions and extractions of water from the rivers that sustain it (e.g., Volga River) instead of the conventionally dominant precipitation/evapotranspiration patterns over the sea surface (Rodell et al., 2018). It is worth mentioning again that the $P - E - R$ equals the changes in TWSAs (TWSC) rather than TWSAs in terms of the water balance equation. Therefore, unlike TWSAs, there are no significant trends in $P - E - R$ over most regions of the world, which is also mentioned by several previous studies (Lv et al., 2019, 2021). Intercomparisons with the GHMs and LSMs further confirm our observation-based evaluations, with relatively fewer magnitudes and significance derived from the substantial uncertainties in simulated $E$ and $R$. In this case, we find an abnormal wetting trend in southwestern America, which might be caused by the severe groundwater pumping and water diversion implicitly considered in the metric $P - E - R$ (Perrone and Jasechko, 2017). Satisfactory consistencies of GHMs and LSMs are also discovered by comparing each subset of $P - E - R$ to the corresponding test results using TWS-DSI. The historical simulations of $P - E - R$ from the ensemble mean of eight GCMs also compare reasonably well with different subsets, though showing the spatial differences over certain regions (e.g., central Europe and south Africa).

Furthermore, we investigate the long-term trends in $P$, $E$, and $R$, respectively, to explain the mechanisms for the changes in land mass wetness/dryness (Figs. S16–S18). Different products and models show consistent spatial patterns for $P$, in which significant ($p < 0.05$) increasing trends are detected in eastern North America (5–10 mm yr$^{-1}$), central Amazon (10–20 mm yr$^{-1}$), northern central and southern Africa (0–5 mm yr$^{-1}$), northern Mediterranean basin (5–10 mm yr$^{-1}$), northwestern China (0–5 mm yr$^{-1}$), eastern Russia (0–5 mm yr$^{-1}$), northern Europe (0–5 mm yr$^{-1}$), and northern Australia (0–10 mm yr$^{-1}$). However, decreasing trends over some areas, including northern Canada ($-5$–$0\,\mathrm{mm\,yr}^{-1}$), southwestern parts of the United States ($-10$ to $-5\,\mathrm{mm\,yr}^{-1}$), central South America ($-15$–$0\,\mathrm{mm\,yr}^{-1}$), Arab regions ($-5$–$0\,\mathrm{mm\,yr}^{-1}$), and northeastern India ($< -20\,\mathrm{mm\,yr}^{-1}$) also exist. In terms of $E$, multiple datasets illustrate generally similar trend distributions with the regional variability in specific areas (e.g., central Africa and Amazon River basin). Significant increases in $E$ are observed over

southern and northern Asia, northern Australia, central and northern Europe, eastern North America, and southern and central northern Africa by all the datasets, with the trends mainly ranging from 0 to 6 mm yr$^{-1}$. This increase might be caused by the warming climate and precipitation changes (Wang et al., 2022). However, we also notice the decreasing trends in the western United States ($-4$–0 mm yr$^{-1}$), central South America ($-8$ to $-4$ mm yr$^{-1}$), and Arab regions ($-2$–0 mm yr$^{-1}$), probably related to the heavy land-cover changes (Ruscica et al., 2022). Moreover, we discover overall similarities among trend estimates in $R$ from different datasets, which are mainly dominated by the precipitation changes regionally with relatively lower amplitudes (roughly between $-12$–12 mm yr$^{-1}$) except for arid central Asia and eastern Europe. In addition, we want to mention that despite the general agreement with different observational products and models, the GCM-based historical trends estimates may have significant uncertainties over some regions, including southern Africa, western America, Amazon, and central Asia (Figs. S16–S18), and hence caution should be taken when interpreting the regional wetting/drying trends in the future scenarios over these regions.

When looking into the respective contributions of $P$, $E$, and $R$ to the changes in $P - E - R$, we find $P$ controls the variations of $P - E - R$ over the majority of the land, including North America, Australia, eastern Russia, northern Europe, and northern Africa. The trends in $P$ over these regions are apparently larger than those of $E$ and $R$, resulting in good agreement with $P - E - R$. Similarly, $E$ governs the changes in $P - E - R$ for southern Africa, northwestern India, southern China, the majority of Europe, and central Russia. It is worth noting that $P$, $E$, and $R$ jointly cause the changes in $P - E - R$ for South America since $P$ and $E/R$ have opposite trends based on the observational products. The Malay Archipelago, including Indonesia and Malaysia, present consistent increasing trends in $P$, $E$, and $R$; thus, the approximately identical contribution of these variables can be attributed. However, it should be noted that the variability of either of these three water balance components (or their combination) may not always translate to the changes in TWSA because human interventions such as reservoir impoundment, water diversion, and groundwater pumping may substantially alter the natural water cycle, as we have discussed previously, taking the Yangtze River basin as an example (e.g., filling of the reservoirs). Although these changes can also be included in the climatic and hydrologic observations in an indirect/implicit way (e.g., increase of $E$ from water impoundment or increase in soil moisture from infiltration), these signals are very difficult to be captured given the considerable uncertainty in different datasets, causing the nonclosure of the water balance (Lehmann et al., 2022). In this case, the assessment of the dryness and wetness from the TWSA perspective becomes more necessary and convincing.

## 3.2 Future projections using ensemble CMIP6 outputs

We project the multimodel ensemble mean trends under different climate change scenarios (SSP126, SSP245, and SSP585) during the future period 2071–2100 using both TWS-DSI and $P - E - R$ (Figs. 1, 2, S14, and S15). Favorably good agreement between TWS-DSI and $P - E - R$ is detected, with the latter presenting a less significant trend, similar to the observations made in previous studies (Lv et al., 2019, 2021). However, we also discover the differences between TWS-DSI and $P - E - R$ over a few high-latitude regions such as northern North America and Russia, which show the wetting trend in $P - E - R$ due to precipitation increase while drying in TWS-DSI probably because of the snowmelt under global warming. GCMs present higher spatial heterogeneity than the historical datasets such as GHMs and LSMs, possibly due to the original coarse spatial resolution of the GCMs and the biases in the models. Specifically, all three scenarios confirm the significant ($p < 0.05$) wetting trends in northern China, southern Mongolia, central Asia, the northern border of Canada, and southern Europe, with the increase in the intensity and spread along with the enhancement of climate scenarios (Figs. 1, 2, S14, and S15). Similarities are found in the drying trends in the majority of Russia, northern North America, and southern Africa. The wetting trends are apparently caused by the increase in precipitation (Fig. S16) (Milly et al., 2005; Seneviratne et al., 2006). The arid Arab region is also projected to become wetter based on TWS-DSI, possibly because of the increase in precipitation. Conversely, the drying trends are mainly controlled by the rapidly intensifying evapotranspiration in a warming climate (Fig. S17) (Allen et al., 2010; Vicente-Serrano et al., 2010), with the precipitation and runoff slightly increasing (Figs. S16 and S18). The obvious drying trend around Canada's subarctic lakes might be related to the high vulnerability to droughts when snow cover declines under increasing temperature (Bouchard et al., 2013). However, there are scenario-variable divergences over the regions of South America, Australia, India, and the Mediterranean basin, which are generally caused by the various patterns in precipitation under different scenarios with the decreasing/increasing evapotranspiration over there. The runoff also follows the patterns of precipitation but with comparably lesser magnitudes.

We conduct a regional study for the QTP as an indicator for global climate change and to demonstrate the temporal changes in the regional dryness and wetness during 1985–2100 (Figs. S19–S20). A significant decrease in the TWSA and the derived TWS-DSI is observed during the reference period 1985–2014 based on different datasets except for the WGHM output. The depletion trend is consistent with previous studies reporting the sublimation/ablation of glaciers and ice caps due to climate warming over decades (Huang et al., 2013, 2021). The drying QTP is also evidenced by the metric $P - E - R$ with a nonsignificant trend based on various

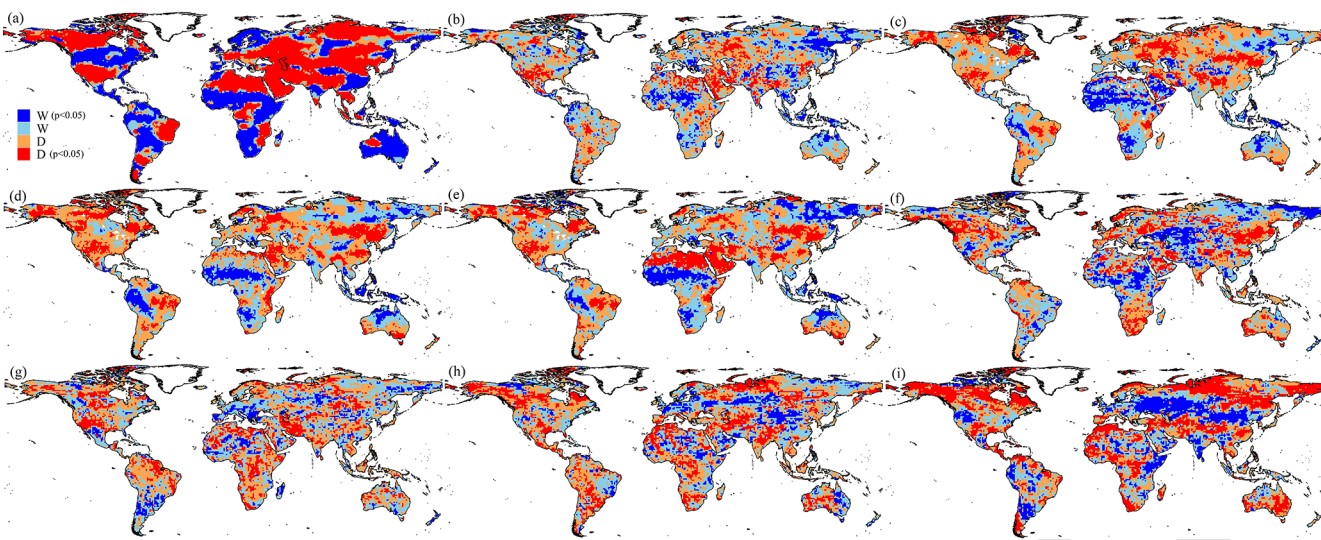

**Figure 1.** Global distribution of the classification in long-term trends in TWS-DSI during **(a–f)** the historical (1985–2014) and future (2071–2100) periods under **(g)** SSP126, **(h)** SSP245, and **(i)** SSP585 scenarios. Note that the historical results are based on the **(a)** GRACE reconstruction, **(b)** WGHM, **(c)** VIC, **(d)** CLSM, **(e)** Noah, and **(f)** ensemble mean of eight GCMs, respectively. The future results are based on the ensemble of eight GCMs. "D" and "W" indicate regions with drying and wetting trends, respectively.

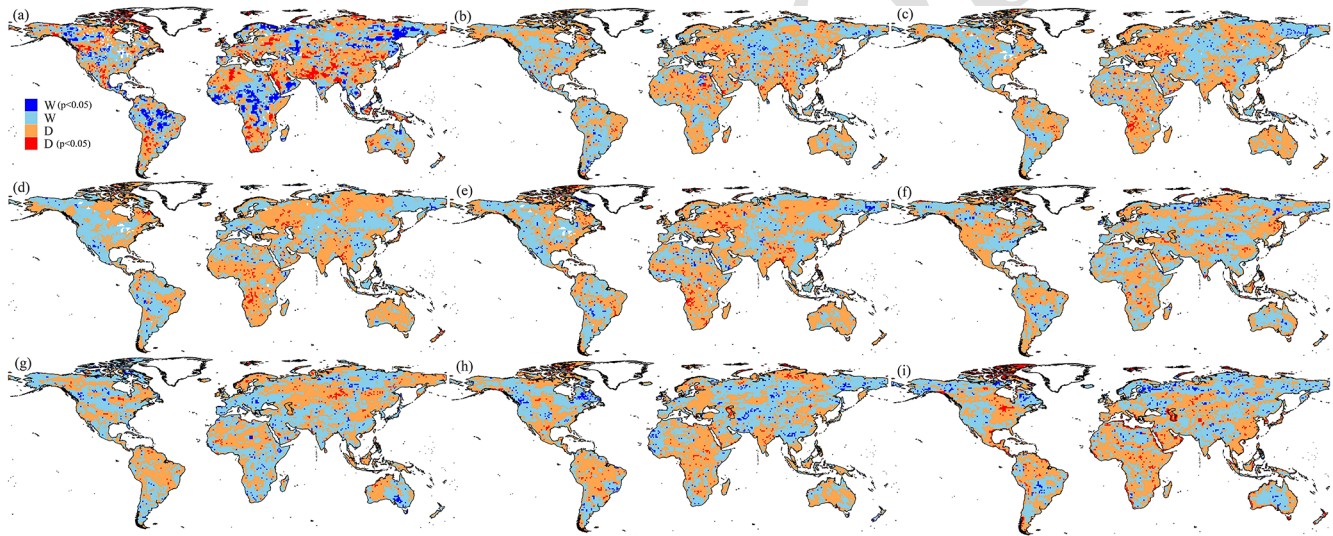

**Figure 2.** Global distribution of the classification in long-term trends in $P - E - R$ during **(a–f)** the historical (1985–2014) and future (2071–2100) periods under **(g)** SSP126, **(h)** SSP245, and **(i)** SSP585 scenarios. Note that the historical results are based on the **(a)** observation-based products (i.e., CRU $P$, GLEAM $E$, and GRUN $R$), **(b)** WGHM, **(c)** VIC, **(d)** CLSM, **(e)** Noah, and **(f)** ensemble mean of eight GCMs, respectively. The future results are based on the ensemble of eight GCMs. "D" and "W" indicate regions with drying and wetting trends, respectively.

datasets, in which both precipitation and evapotranspiration increase. In addition, the QTP is expected to undergo continuous drying trends based on TWSA and TWS-DSI stemming from a warming climate, which can be more intensive under higher climate scenarios from SSP245 and SSP585 conditions (Fig. S19). Similarly, regional precipitation and evapotranspiration also show increasing patterns, with the runoff generally unchanged (except during the end of the 21st century under the SSP585 scenario). However, the variable $P - E - R$ does not present decreasing trends like the TWSA (and TWS-DSI). The differences might be attributable to the biases in the projected evapotranspiration and runoff, which might underestimate some key components such as an increase in sublimation and surface runoff due to warming-induced melt of ice, snow, and glaciers. Despite this, it is worth noting that the modeled TWS-DSI-based evaluation

can also overestimate the true trend of the land mass because the important surface water is not physically considered in several models (e.g., Noah), especially in the context of significantly growing lake volume over the QTP (Zhang et al., 2021).

## 3.3 Assessment of the DDWW paradigm

Combined with the climate regions classified by AI, we further test the DDWW paradigm at a 5 % significance level using both TWS-DSI and $P-E-R$ over global land in the past and future (Figs. 3 and 4). We observe apparent consistency in the spatial distribution of the test results based on different indices except for the high-latitude regions under future projections, in line with the long-term trend estimations, while the land area having significant patterns from TWS-DSI is more than that from $P-E-R$ as investigated previously. In addition, different datasets (e.g., GHMs and LSMs) produce reasonably consistent spatial distributions except for the regional variabilities over certain regions such as North Africa. We also note that relatively larger biases could occur in several regions including the western United States and central Asia, highlighting the uncertainties in the future projections based on the CMIP6 GCMs. As reported in Table S3, limited proportions ($< 10$ %) of area illustrating the "transition gets drier" (TD) and "transition gets wetter" (TW) patterns are estimated in both past and future periods. Much of the land area over the Arab regions, eastern Asia, and the southwestern United States shows the "dry gets drier" (DD) phenomenon. In contrast to that, a substantial portion of area over the arid regions of northern and southern Africa, Australia, and central Asia shows the "dry gets wetter" (DW) hypothesis. Moreover, the "wet gets wetter" (WW) paradigm is mainly confirmed in eastern Russia, northern Amazon, southern China, and the eastern United States, with the "wet gets drier" (WD) pattern happening in central Africa, eastern Amazon, middle Europe, western Canada, and northern Asia. The differences between test results from TWS-DSI and $P-E-R$ are mainly in southern China and lands north of the Caspian Sea, which are caused by the divergent meanings in the metrics. For example, a significant increase in $E$ over southern China is shown as the drying trends of $P-E-R$, instead of the wetting trends of TWS-DSI induced by the extensive reservoir impoundment (e.g., Three Gorges Dam). The differences are highlighted by the future projections over high-latitude regions such as northern Russia and North America as well as central Africa, especially under the SSP585 scenario. Despite this, a similar pattern revealed by both variables under the SSP126 scenario shows the continued tendency when compared with the historical results (Figs. 3 and 4). However, some regions like southern Europe and southeastern South America present strong wetting trends due to an increase in precipitation (Coppola et al., 2021); the opposite changes are discovered over northern South America. Nevertheless, the SSP245 scenario presents

a slightly different distribution from historical results, with many regions in northern and central Asia and central Europe showing DW and WW situations instead of DD and WD. In addition to that, the southern and northwestern parts of China, together with the majority of Russia, show the WD situation, while the DD paradigm is gradually dominating Australia. This difference is further confirmed based on the results under the SSP585 scenario (Figs. 3 and 4). These results correspond with the climatic and hydrologic fluxes such as $P$, $E$, and $R$ as well as their residuals ($P-E-R$), indicating the consistency between the atmospheric and terrestrial conditions under climate change.

Global statistics of the regions with various patterns during the historical (1985–2014) and future (2071–2100) periods are shown in Fig. 5. During the 1985–2014 period, a percentage of as high as 82.8 % of the land area shows significant trends in either wetting or drying ($p < 0.05$) based on the GRACE reconstruction. Further, 40.84 % of the area shows the DDWW paradigm, in which 20.17 % and 20.67 % of the area is drying and wetting, respectively; 35.43 % of the area, however, shows the opposite pattern of DW (16.13 %) and WD (19.30 %), respectively. The percentages of the global land supporting/opposing the DDWW paradigm from the GHMs and LSMs are relatively lower than those from the GRACE reconstruction using TWS-DSI, which are reflected by the fewer proportions with significant trends. For example, the percentage of the land area showing the DDWW paradigm ranges from 11.01 % (VIC) to 18.95 % (Noah) and from 10.21 % (WGHM) to 16.4 % (VIC) for the opposite pattern. The test results based on $P-E-R$ indicate a similar mismatch of the DDWW paradigm with 12.54 % and 6.62 % of the land area validating and combating the DDWW paradigm, respectively, based on the observational products (Fig. S21 and Table S4). Nevertheless, GHMs and LSMs report nonsignificant trends ($p > 0.05$) over more than 90 % of land area. In short, the confirmed percentage for the DDWW paradigm (11.01 % to 40.84 %) for the land mass (represented by TWS-DSI) in our study is higher than that for the land surface (represented by precipitation, evaporation, and aridity) in a previous study (10.8 %) (Greve et al., 2014). Feng and Zhang (2015) used soil moisture to conclude that a proportion of 15.12 % followed the DDWW pattern, while a percentage of 7.7 % of the land showed an opposite pattern between 1979 and 2013, which is relatively lower than our study. Yang et al. (2019) applied a combined measure employing six different drought indices to evaluate the DDWW paradigm and discovered the percentage following and opposing the DDWW paradigm is 29 % and 20 %, respectively, during the 1982–2012 period, typically consistent with our study. Chang et al. (2020) utilized the GRACE data during 2002–2017 and reported that the area having the DDWW pattern reached 11.2 % except for 4.7 % of cold regions over global land, which is comparatively lower than our study. Observed differences among various studies are attributed to

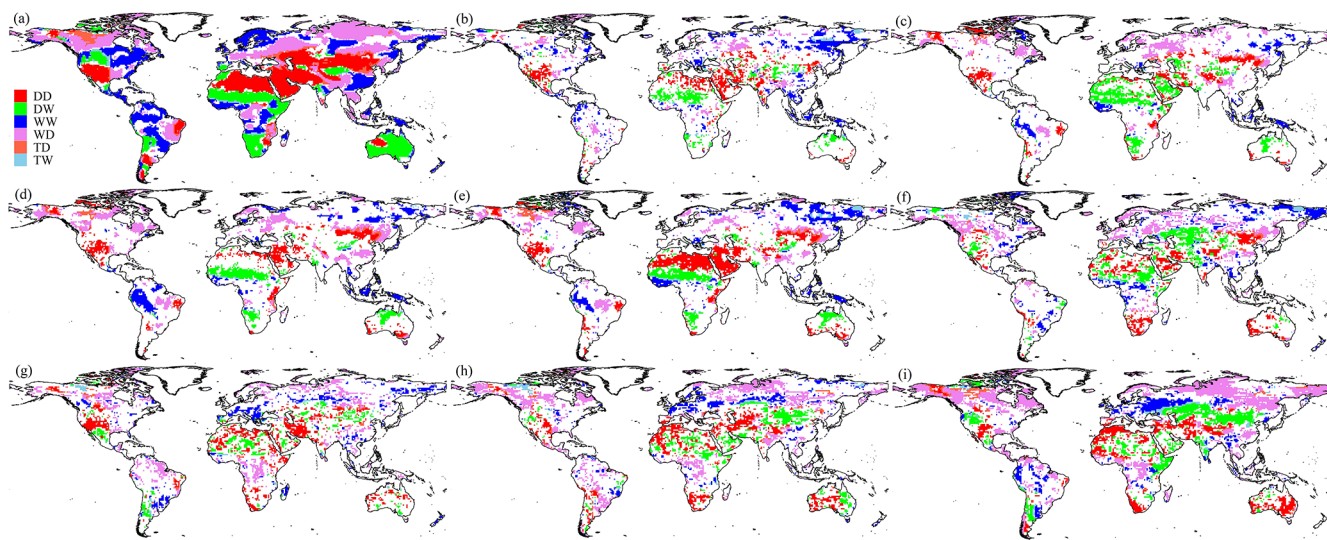

**Figure 3.** Global assessment of the DDWW paradigm based on TWS-DSI during the **(a–f)** historical (1985–2014) and **(g–i)** future (2071–2100) periods under **(g)** SSP126, **(h)** SSP245, and **(i)** SSP585 scenarios. Note that the historical results are based on the **(a)** GRACE reconstruction, **(b)** WGHM, **(c)** VIC, **(d)** CLSM, **(e)** Noah, and **(f)** ensemble mean of eight GCMs, respectively. The future results are based on the ensemble of eight GCMs. DD indicates the dry gets drier, DW indicates the dry gets wetter, WW indicates the wet gets wetter, WD indicates the wet gets drier, TD indicates the transition gets drier, and TW indicates the transition gets wetter.

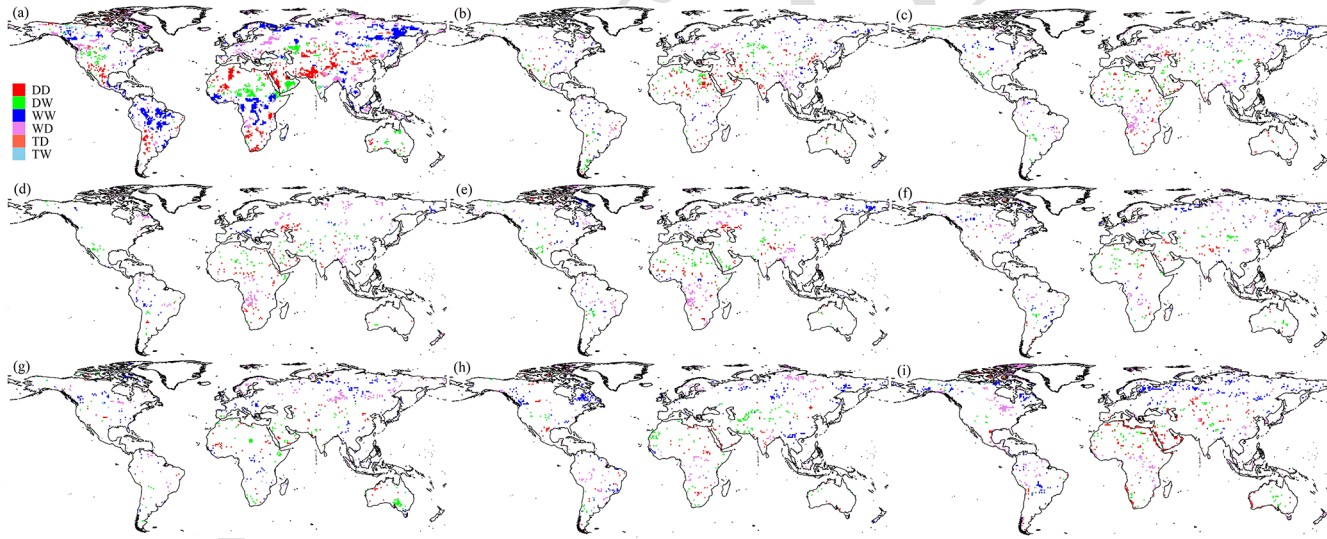

**Figure 4.** Global assessment of the DDWW paradigm based on $P - E - R$ during the **(a–f)** historical (1985–2014) and future (2071–2100) periods under **(g)** SSP126, **(h)** SSP245, and **(i)** SSP585 scenarios. Note that the historical results are based on the **(a)** observation-based products (i.e., CRU $P$, GLEAM $E$, and GRUN $R$), **(b)** WGHM, **(c)** VIC, **(d)** CLSM, **(e)** Noah, and **(f)** ensemble mean of eight GCMs, respectively. The future results are based on the ensemble of eight GCMs. DD indicates the dry gets drier, DW indicates the dry gets wetter, WW indicates the wet gets wetter, WD indicates the wet gets drier, TD indicates the transition gets drier, and TW indicates the transition gets wetter.

the differences in datasets used, metrics employed for assessment and their governing mechanisms, and the study period.

In climate model projections, the proportion of areas supporting the DDWW paradigm is 14.66 %, 14.26 %, and 17.08 % under SSP126, SSP245, and SSP585 scenarios, respectively, for TWS-DSI. Alternatively, the fraction of the global land area having the opposite DDWW pattern achieves 13.84 %, 18.72 %, and 26.64 %, respectively. The percentage of areas with significant wetting and drying trends slightly increases over the enhancement of emission scenarios, consistent with the increase of DDWW-validated areas from SSP126 to SSP585 scenarios (Figs. 3 and 4). The evalu-

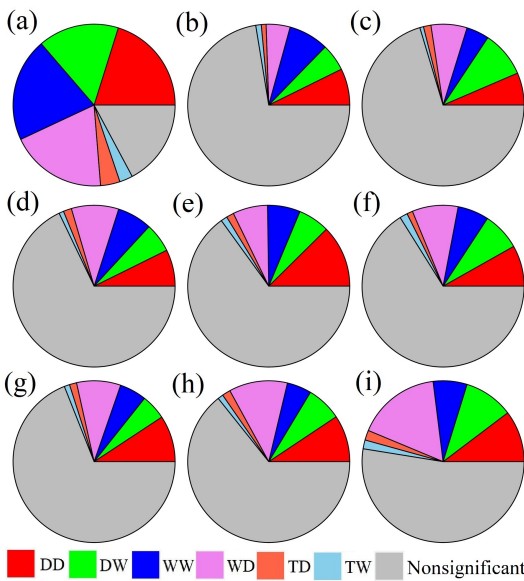

**Figure 5.** Fraction of the global land area (in percentage) with different patterns during the **(a–f)** historical (1985–2014) and **(g–i)** future (2071–2100) periods under **(g)** SSP126, **(h)** SSP245, and **(i)** SSP585 scenarios based on TWS-DSI. Note that the historical results are based on the (a) GRACE reconstruction, **(b)** WGHM, **(c)** VIC, **(d)** CLSM, **(e)** Noah, and **(f)** ensemble mean of eight GCMs, respectively. The future results are based on the ensemble of eight GCMs. DD indicates the dry gets drier, DW indicates the dry gets wetter, WW indicates the wet gets wetter, WD indicates the wet gets drier, TD indicates the transition gets drier, and TW indicates the transition gets wetter. Nonsignificant indicates the regions showing nonsignificant ($p > 0.05$) trends in TWS-DSI.

ation results from the perspective of $P - E - R$ are generally lower than 5 % because of the nonsignificant trends in the variable, highlighting the unsupported DDWW paradigm in this regard. However, as we have mentioned previously, the internal variability of climate models might affect the potential agreement with the DDWW pattern (Kumar et al., 2015), which is also reflected by the differences between the GCMs and different models/products during the historical period (Tables S3–S4). Greve and Seneviratne (2015) used climate projections from CMIP5 to establish the measure $P - E$ for the assessment of the DDWW paradigm and discovered the hypothesis was validated over 19.5 % of land area between 2080 and 2100 under the RCP8.5 scenario, which is close to our result (17.08 %). Moreover, Y. Li et al. (2021) further applied the $P - E$ index to test the DDWW theory based on GCMs from the third phase of Paleoclimate Modelling Inter-comparison Project (PMIP3) simulations, concluding a similar proportion of 22.8 % of the global land to our study that reflected the DDWW paradigm. This similarity reveals the consistent terrestrial responses to the atmospheric variations under future warming for both metrics.

## 3.4 Uncertainties, implications, and way forward

Each ensemble member of the datasets used in this study has embedded uncertainties inherently originating from one or more forcing variables, simplified assumptions of complex processes in the models and their physical structure, retrieval algorithms, and systematic biases, which might have inevitably propagated to the results presented herein. For example, the original GRACE mascon observations contain the measurement error and signal leakage at the gridded scale, which persists in the reconstruction of TWSA when training via statistical methods (F. Li et al., 2021). Unlike observed GRACE and reconstructed GRACE-like data, simulations from the models (GHMs, LSMs, and GCMs) are inherently featured by incomplete TWSA representation (Table S1). They are generally based on simplified hydrological processes, resulting in the lack of certain TWSA components. For example, the widely used Noah and VIC models lack surface water and groundwater storage in TWSA (Scanlon et al., 2018). Similarly, GCMs can only simulate the snow water and soil moisture within a limited depth from 2 to 10 m below the land surface (Xiong et al., 2022a). This inadequate representation of TWSA (and hence TWS-DSI) in these global models can lead to regional bias in some aquifers with overexploitation of the particular TWSA components (e.g., groundwater depletion in North China Plain) and therefore should be cautioned, especially dealing with the seasonal analyses. Overall, the models with completed TWS components are more suitable for assessing the TWSA changes at the global scale for future research, such as the continuously developing hyper-resolution global hydrological models (e.g., WGHM), which can help to avoid the uncertainty associated with the lack of key TWSA elements in most LSMs (e.g., surface water and groundwater) (Pokhrel et al., 2021).

Moreover, the eight CMIP6 GCMs are forced with the future projections of many meteorological variables such as precipitation and air temperature, which have been reported to show variable-specific biases over the global land (Eyring et al., 2016; Kim et al., 2020). Despite employing bias correction with GRACE data, uncertainty from the forcing and models can influence the accuracy of TWSA simulations (Xiong et al., 2022a). Advanced bias-correction methods (e.g., Lange, 2019; François et al., 2020) might play critical roles in reducing such errors in meteorological variables for future hydrologic impact studies, especially when combined with the start-of-the-art GHMs and LSMs as mentioned above. The inclusion of more GCMs can also help to estimate the uncertainties in the meteorological inputs in climate change scenarios. Although it is challenging to explicitly attribute and quantify these uncertainties in the absence of a "true" reference observation dataset, the ensemble averaging method has been used to integrate the multisource TWSA data. Moreover, since the meaning, and hence the results and interpretation of "dry" and "wet", varies across dis-

ciplines, land or ocean, target variable(s), and the problem in question (Roth et al., 2021), future studies may focus on various spatial (e.g., local, regional, basin, and zonal averages) and temporal (monthly, seasonal, and annual) scales using our processed data with additional model outputs (e.g., more GCMs).

To investigate the influence of different models on the robustness of the evaluation for the DDWW paradigm, we carry out an independent analysis at the individual member level during the future period 2071–2100 (see Fig. S22). We find the differences among different members of the CMIP6 archive. The GFDL-ESM4 and MIROC6 models present overestimations, but the IPSL-CM6A and CanESM5 models underestimate different percentages compared with the ensemble mean. Specifically, the area dominated by the DDWW paradigm changes from 8.16 % (ACCESS-ESM1-5) to 19.36 % (MIROC6), while that showing the opposite pattern ranges from 7.33 % (CanESM5) to 14.57 % (MPI-ESM1-2-HR) under the SSP126 scenario. For the SSP245 scenario, the DDWW-validated regions account for 6.98 % (CanESM5) to 18.54 % (GFDL-ESM4); the opposite pattern occurs over a range from 8.71 % (CanESM5) to 12.64 % (MPI-ESM1-2-HR) of land. The proportion supporting the DDWW paradigm varies from 9.71 % (CanESM5) to 20.08 % (GFDL-ESM4), while that presenting the opposite pattern ranges from 8.19 % (MPI-ESM1-2-LR) to 18.68 % (ACCESS-CM2) under the SSP585 scenario. Overall, the comparatively large difference among various models might source from unforced internal climate variability of distinctive CMIP6 members and different emission scenarios (Kumar et al., 2015).

Our choice of the significance level (i.e., 0.05) may also affect the rationale of the DDWW examination results. Therefore, different significance levels are alternatively tested (see Figs. S23–S24 and Tables S5–S6). At a significance level of 0.01, a decrease in 3.21 % (37.63 %) of the land area agreeing well with the DDWW theory is detected, with a reduction of 2.65 % (32.78 %) in area illustrating the opposite pattern during the 1985–2014 period for the GRACE reconstruction. Similar decreases in the proportion of the DDWW-dominated area ranging from 5.19 % (SSP245) to 7.2 % (CLSM) are also discovered in the GHMs, LSMs, and GCMs. As for the 0.1 significance level, the DDWW-validated regions account for 42.49 % (+1.65 %) of the total area, with 36.89 % (+1.46 %) of land agreeing with the opposite hypothesis compared to those at the 0.05 level. In the future period, a similar pattern is discovered that both DDWW-confirmed and DDWW-opposed regions are increasing on account of the enhancement of projected strength of radiative forcing, with the reduction of the area showing nonsignificant trends in wetting and drying. However, the magnitudes of results at the 0.01 significance level are generally lower than those at the 0.1 significance level due to the different thresholds of the detected trends in drying and wetting. Considering the similar tendency with marginal effects of the varying choices of

the $p$ value (e.g., 4.86 % change in DDWW area from 0.01 to 0.1 level for the GRACE reconstruction during 1985–2014), our adopted significance level (i.e., 0.05) can reasonably and robustly explain the global trends of dryness and wetness. Given the inherent magnitude bias from various GCMs projections, the ensemble averaging method has the potential to provide alternative estimates over data-sparse areas globally like Africa and central Asia.

Despite the multisource uncertainties, our study provides important implications for the long-term trends in dryness and wetness of the global land mass in the past and future from the perspective of TWSA. Compared with other widely used indices that are purely derived from hydrometeorological variables (e.g., SPI, SPEI, and PDSI (Palmer Drought Severity Index)) or incorporate a single component of the TWSA (e.g., SSI, SGI, and SRI), our developed TWS-DSI is able to describe the overall status of the land system, which is jointly influenced by different components including soil moisture, river runoff, and groundwater that play different roles in the hydrological cycle (Tapley et al., 2019). Although other indices may undoubtedly perform similarly for the specific variable in question, they tend to present equivocal inferences for the total water storage. It can be easily understood by the example of soil moisture or evapotranspiration-based indices in a highly irrigated area such as the Ganges River basin. TWS is unremittingly declining due to the overexploitation of groundwater for agriculture in this region (Rodell et al., 2009), while $E$ or soil moisture may have positive trends, thus attenuating the actual TWS situation. Moreover, the adopted TWS-DSI is suitable and feasible for comparing dryness and wetness status for different locations and periods (Zhao et al., 2017). Furthermore, the projected changes in the global TWSA and associated TWS-DSI improve our understanding of the large-scale hydrological response to climate change, particularly in regions with strong human interventions, such as the south and east of Asia.

## 4 Conclusion

This study performs a global examination for the dry gets drier, wet gets wetter paradigm from a terrestrial water storage perspective in the past and future. The historical TWS-DSI monthly time series over global land during 1985–2014 is calculated from two GHMs (VIC and WGHM), two LSMs (Noah and CLSM), and one GRACE reconstruction. In addition, future projections of TWS-DSI from 2071 to 2100 under SSP126, SSP245, and SSP585 scenarios are derived from the average of eight selected CMIP6 GCMs after bias correction using GRACE observations. Further, the DDWW paradigm has been evaluated with a significance level of 0.05 from the perspective of terrestrial water storage change. We also establish the metric $P - E - R$ based on multiple observational products and from the same models as the TWS-DSI for comparison. The uncertainty sourced from different

choices of models, methods, and confidence levels has been discussed systematically. The new findings are summarized as follows.

1. During the historical period, the percentage of global land area presenting significant ($p < 0.05$) drying and wetting trends ranges from 13.06 % (WGHM) to 43.35 % (GRACE reconstruction) and 13.7 % (CLSM) to 39.43 % (GRACE reconstruction), respectively. The wetting trends are mainly in northern Australia, northern and southern Africa, southern and northwestern China, western South America, the central United States, and eastern Russia, while drying trends are found in the Arab region, western Brazil, northeastern Asia, and the South and North American continent. During the future period under climate change, the proportion of drying areas (always ∼ 10 % higher than wetting) with a significant slope increases from the SSP126 (19.52 %) to SSP585 (29.04 %) scenario. A similar change is detected in the percentage with significant wetting trends, which reaches 11.48 %, 13.01 %, and 18.42 % under SSP126, SSP245, and SSP585 scenarios, respectively.

2. A total of 11.01 % (VIC) to 40.84 % (GRACE reconstruction) of the global land area shows the DDWW paradigm valid, in which the drying and wetting area account for 6.47 % (VIC) to 20.17 % (GRACE reconstruction) and 4.54 % (VIC) to 20.67 % (GRACE reconstruction), respectively, during the 1985-2014 period. However, the area showing opposite patterns, like "dry gets wetter" (DW) or "wet gets drier" (WD), account for 10.21 % (WGHM) to 35.43 % (GRACE reconstruction) of the global land, respectively. The proportion of areas supporting (opposing) the DDWW paradigm is 14.66 % (16.76 %), 14.26 % (18.72 %), and 17.08 % (26.64 %) under SSP126, SSP245, and SSP585 scenarios, respectively. Regional assessment for the QTP reveals the drying trends of the land mass primarily attributable to the sublimation/ablation of glaciers and ice caps, together with a continued tendency in future warming climates until the end of the 21st century.

3. Sensitivity analysis on different choices of significance levels from 0.01 to 0.1 for the long-term trends indicates similar patterns, in which the maximum decrease (increase) in the DDWW-validated regions reaches −7.4 % (4.47 %) historically under the 0.01 (0.1) level, respectively. Such consistency is also evidenced by the projected TWS-DSI in the future under various scenarios. Moreover, independent experiments based on the individual TWSA datasets suggest that the divergent data sources might lead to model-variable biases for both the DDWW-agreed and DDWW-opposed patterns. The use of distinctive GCMs also suggests slightly overrated (e.g., GFDL-ESM4) and underrated (e.g., CanESM5)

percentages of such patterns in the future under multiple emission scenarios.

New insights from the TWSA perspective highlight that the widely used DDWW paradigm is still challenged in both historical and future periods under climate change. The differences between test results based on $P - E - R$ imply the robustness of our developed TWS-DSI in capturing the total land water variations induced by climate change and human activities, suggesting potentially new knowledge in the land hydrology field.

*Data availability.* The data used in this study are open-access and publicly available: GRACE solution (https://www2.csr.utexas.edu/grace/RL06_mascons.html, GRACE, 2022), GRACE reconstruction (https://doi.org/10.5061/dryad.z612jm6bt, Li, 2021), GHMs (WGHM, Müller Schmied et al., 2021; VIC, TS3 https://doi.org/10.5067/ZRIHVF29X43C, Beaudoing and Rodell, 2020), LSMs (Noah, https://doi.org/10.5067/QN80TO7ZHFJZ, Beaudoing and Rodell, 2019; https://ldas.gsfc.nasa.gov/gldas), GCMs (https://doi.org/10.5067/SGSL3LNKGJWW, Li et al., 2020), and climatic and hydrologic datasets (precipitation and potential evapotranspiration, https://crudata.uea.ac.uk/cru/data/hrg/cru_ts_4.06/TS4; runoff, https://doi.org/10.1029/2020WR028787; evapotranspiration; https://www.gleam.eu/TS5). The data used for deriving figures in this study have been made publicly available via the Zenodo platform (https://doi.org/10.5281/zenodo.7212993, Xiong et al., 2022).

*Supplement.* The supplement related to this article is available online at: https://doi.org/10.5194/hess-26-1-2022-supplement.

*Author contributions.* JX conceived and designed the experiments. JX performed the experiments. JX and A analyzed the data. JX, SG, A, JC, and JY wrote and edited the paper.

*Competing interests.* The contact author has declared that none of the authors has any competing interests.

*Acknowledgements.* The numerical calculations in this paper were done on the supercomputing system in the Supercomputing Center of Wuhan University.

*Financial support.* This research has been supported by the National Key Research and Development Program of China (grant no. 2021YFC3200303) and the National Natural Science Foundation of China (grant no. U20A20317).

*Review statement.* This paper was edited by Adriaan J. (Ryan) Teuling and reviewed by Yannis Markonis and two anonymous referees.

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

TS4     We still need a reference for this link.

TS5     We still need a reference for this link.