# Peer review of "Global evaluation of the dry gets drier and wet gets wetter paradigm"

_Hydrology and Earth System Sciences, 2022_

## Author Response (AR1)

**Cover Letter**

Dear Editor,

We have substantially revised our manuscript according to the reviewer's insightful comments and suggestions. We are pleased to resubmit the revised version of our manuscript titled "*Global evaluation of the dry gets drier and wet gets wetter paradigm from terrestrial water storage changes perspective*" (#ID: **hess-2022-190**).

All the comments are addressed in the new version of the manuscript, and several main and supplementary figures, tables, and discussions have been added to enrich the study. Below is the attached point-by-point explanation of our correspondence for each comment or suggestion by the reviewers. All additional and changed parts of the text (except some minor language corrections) are marked in BLUE for easy review.

We sincerely hope you and the reviewers will find the revised version of the manuscript much more comprehensive and robust. All the authors have reviewed the manuscript and agree to the submission of the manuscript. We look forward to hearing from you.

Thank you for your time and efforts on our manuscript again.

Yours sincerely,

August 2$^{nd}$, 2022
Prof. Shenglian Guo
Corresponding author
State Key Laboratory of Water Resources and Hydropower Engineering Science,
Wuhan University, Wuhan 430072, P. R China
**E-mail:** slguo@whu.edu.cn

**Reply to Reviewers' comments (Reviewer#1)**

**Legend**

Reviewers' comments

Authors' responses

Direct quotes from the revised manuscript

We thank the reviewer for his/her time in reading our manuscript and detailed comments on our manuscript. Point-by-point replies to the comments or suggestions made can be found below. Overall, we have made the following major changes to the manuscript:

- Performed additional analysis using P-E-R and analyzed and compared the results with TWSA-DSI.
- Instead of showing only the ensemble mean of various model and observation-based results, we have now shown the results from individual datasets during the historical period (1985-2014).
- Added extensive discussion about the various mechanisms and governing processes for the observed patterns and the similarities and disparities from the previous studies.

**Reviewer #1:** This is a resubmitted manuscript and this is my second review. This study re-examine the "dry gets drier and wet gets wetter" (DDWW) paradigm using the terrestrial water storage anomaly (TWSA) derived from GRACE observational products, land surface models, and GCMs. The results showed the global patterns of dryness/wetness trends in both history (1985-2014) and future (2071-2100).

In this version, the authors have improved the text, added discussion, and provided more uncertainty analyses. I am happy with the authors' efforts. However, there are substantial issues which need to be addressed. The authors should set out to solve scientific problem rather than analyzing data. At present, I did not feel the new knowledge and new (and convincing) methods provided by this paper. At least, the authors have not fully express the innovation and significance of this study.

Response: Thank you very much for the second review of our manuscript and for encouraging feedback. We provide the global evaluation of the topical DDWW paradigm from the TWSA perspective in the past and future, which has never been performed before. Furthermore, we have added mechanism analysis of the TWSA patterns in the revised version, including comparison with the conventionally used wetness/dryness metric P-E-R and performing regional studies with significantly decreasing TWS-DSI over the Qinghai-Tibetan Plateau. New knowledge shows the contradictory DDWW patterns in terrestrial land mass over global land for the past and future based on a total of 18 datasets, including LSMs, GHMs, GCMs, and observation-based products. We have highlighted the additional insights and have clarified the innovation and significance of this study in the new version.

Concept: The authors should recall the original meaning of "dry gets drier and wet gets wetter" paradigm from existing studies, because the title/the authors intend to perform a "re-examine" work. I think the authors acknowledge that the DDWW rule is used to explain the changing trend of surface dryness/wetness or climate condition, while this study explains the DDWW rule from a TWS perspective which includes groundwater/glacier changes. The GRACE observation contains the signal of changes in groundwater/glacier. As the climate warms, ice/glaciers are degrading with an increase in runoff/soil moisture (moisten the land surface). Meanwhile, as the mass decreases (water flows away), what GRACE observes is a decrease trend in gravity (drying). There are processes in the opposite direction. As such, the TWSA trends can be opposite with the previous studies focusing on the land-surface conditions (soil moisture/runoff and ET) (Wang et al., 2021: Long-term relative decline in evapotranspiration with increasing runoff on fractional land surfaces; Yang et al., 2019: Combined use of multiple drought indices for global assessment of dry gets drier and wet gets wetter paradigm). Rather than a new perspective, I would also think of this study as a simulation or an application of GRACE, land surface models, and climate models.

Response: We thank the reviewer for this important suggestion about the title.
Title Change: Since we provide the first global evaluation of the DDWW paradigm from the TWSA perspective in both the past and future, we have updated the title to - 'Global evaluation of the dry gets drier and wet gets wetter paradigm from terrestrial water storage changes perspective' to better reflect the approach and contents of the manuscript.

Importance of and need for TWSA perspective: We agree that conventionally DDWW paradigm has been studied either by directly using the two competing variables, i.e., precipitation and evapotranspiration (Held and Soden, 2006), or the derived indices such as P/ET (Greve et al., 2014), SPEI (Yang et al., 2019), and PDSI (Hu et al., 2019). However, there has been increasing attention to the DDWW paradigm from different perspectives (e.g., soil moisture (Feng and Zhang, 2015) and runoff (Yang et al., 2019)) in the last decade. Inconsistent usage of the term "wetter" and "drier" across disciplines (Roth et al., 2021) and different physical meanings of these variables further limit their implications in the context of total land water storage. Since the terrestrial/land water storage (i.e., TWS) is a crucial variable for the community working on, e.g., ecosystem functioning (Humphrey et al., 2018), sea-level budget studies (Frederikse et al., 2018), terrestrial water balance, hydroclimatic extremes, and freshwater availability (Rodell et al., 2018), it merits indispensable consideration. Given the different meaning of TWSA with previous metrics (e.g., P and ET), the evaluation of the DDWW paradigm from the TWSA perspective and inter-comparison and subsequent analysis of governing processes/mechanisms as carried out in our study can potentially provide new evidence. Please also see our responses to the major comments on the 'methods' below and response to specific comment#4 for a detailed explanation of the similarity, differences, and significance of TWSA compared to P/E (or derived indices).

As rightly indicated by the reviewer, unlike many evaluations focusing on the

land surface water balance (e.g., P-E), our developed TWS-DSI contains the signals of groundwater/glacier that were impossible to be considered previously. We regret and are surprised that the reviewer has an impression of the revised manuscript (with, in our understanding, all previously raised concerns resolved) different than that of the original manuscript ('The topic is interesting and this study potentially provides a new perspective.'). However, we have changed the title and have tried to thoroughly incorporate all the suggestions in this version.

References:

Feng, H., Zhang, M., 2015. Global land moisture trends: drier in dry and wetter in wet over land. Sci. Rep. 5, 18018. https://doi.org/10.1038/srep18018

Frederikse, T., Jevrejeva, S., Riva, R. E. M., and Dangendorf, S.: A Consistent Sea-Level Reconstruction and Its Budget on Basin and Global Scales over 1958–2014, J. Climate, 31, 1267–1280, https://doi.org/10.1175/jcli-d-17-0502.1, 2018.

Held, I. M., Soden, B. J. 2006. Robust responses of the hydrological cycle to global warming. Journal of climate, 19(21), 5686-5699.

Humphrey, V., Zscheischler, J., Ciais, P. et al. 2018. Sensitivity of atmospheric CO2 growth rate to observed changes in terrestrial water storage. Nature 560, 628–631. https://doi.org/10.1038/s41586-018-0424-4

Hu, Z.Y., Chen, X., Chen, D.L. Li, J.F., Wang, S., Zhou, Q., Yin, G., Guo, M. 2019. "Dry gets drier, wet gets wetter": a case study over the arid regions of central Asia. Int J Climatol 39(2):1072–1091

Rodell, M., Famiglietti, J.S., Wiese, D.N. et al. 2018. Emerging trends in global freshwater availability. Nature 557, 651–659. https://doi.org/10.1038/s41586-018-0123-1

Roth, N., Jaramillo, F., Wang-Erlandsson, L., Zamora, D., Palomino-Ángel, S., Cousins, S. A. 2021. A call for consistency with the terms 'wetter'and 'drier'in climate change studies. Environmental Evidence, 10(1), 1-7.

Yang, T., Ding, J., Liu, D., Wang, X., Wang, T., 2019. Combined Use of Multiple Drought Indices for Global Assessment of Dry Gets Drier and Wet Gets Wetter Paradigm. J. Clim. 32, 737–748. https://doi.org/10.1175/JCLI-D-18-0261.1

Method: In the discussion, the authors need to justify why it is necessary to assess the changes in dryness/wetness from a perspective of terrestrial water storage change? What are the advantages of the methodology used in this method? This study has many redundant operations (e.g., the use of GRACE to correct GCMs). I feel if the study directly using P-ET is more convincing than using partial outputs (soil moisture, snow water...). While changes in TWSA do not equal to changes in surface dryness/wetness, there should have "bridges" to connect the integrated TWS and various land-surface processes (runoff, soil moisture and ET) (Trautmann et al, 2022: The importance of vegetation in understanding terrestrial water storage variations). It is a pity that this study did not find such "bridges" as it leaned toward analyzing data. The use of TWS retrieved by the GRACE to correct GCM simulations is not convincing. Not only are there many uncertainties in the GRACE retrieval product, but also what GRACE observes is completely different from what GCMs simulate (Table S2). Since these models express different objects, how can these outputs ensemble? What will happen if the study does not use GRACE to correct the GCM simulations as most climatologists do? There are still have a prediction result from GCM, right? What are the differences? One way is to show that the corrected results are more reliable than the previous one, which may involve using in-situ observed data. Moreover, the authors criticize the use of P-ET as an indicator to identify dry/wet changes, but I think P-ET is closer to changes in TWS because various hydrological models and GCMs appear to do not account for surface water storage. I suggest the authors provide a technical route.

Response: We thank the reviewer for the comment. Please find the detailed explanation of all the concerns below.

Significance of TWSA and performance disparity:

Let us briefly discuss the significance of TWS using two examples from a process perspective, i.e., ecosystem functioning (Humphrey et al., 2018) and freshwater availability (Rodell et al., 2018). The inter annual fluctuations of TWSA significantly influence the terrestrial carbon sink and are essential for the global water and carbon cycles-two major cycles of the earth system sciences (Humphrey et al., 2018). Its long-term trends are also indicative of the global water's landscape influenced by climate variability, climate change, and human activities and offer important inferences for global water and food security (Rodell et al., 2018). Although the amount of water stored in land is governed by the precipitation (and evapotranspiration and runoff) influxes (out fluxes), the change in the storage is governed by the synergistic impact of climatic and human-induced changes, which are imperative for a wide range of the applications. Hence, we infer that water storage is more relevant than P or ET or a combination thereof. Please also see our response to the previous comment for more details on the need and importance of the TWSA perspective.

References:
Humphrey, V., Zscheischler, J., Ciais, P. et al. 2018. Sensitivity of atmospheric CO2 growth rate to observed changes in terrestrial water storage. Nature, 560, 628–631. https://doi.org/10.1038/s41586-018-0424-4
Rodell, M., Famiglietti, J.S., Wiese, D.N. et al. 2018. Emerging trends in global freshwater availability. Nature, 557, 651–659. https://doi.org/10.1038/s41586-018-0123-1

Advantages of the methods used:

Our study establishes the normalized TWS-DSI index based on different TWSA datasets, which accounts for the regional hydro climatological variability and is suitable for comparing dryness/wetness status for different locations and periods (Zhao et al., 2017). In addition, the modified Mann-Kendall test used for the long-term trend estimations could avoid the autocorrelation of the time series (Hamed and Rao, 1998). The future projections from CMIP6 GCMs are bias-corrected using the trend-preserving method (Hempel et al., 2013). We fully agree the GCM simulations have considerable uncertainties, which might be further strengthened over regions with significant variations in vegetation, surface water, and groundwater due to the constrained representations of TWS in GCMs. To show the difference, we selected two typical regions (i.e., Amazon and Mekong River basins) with abundant surface and groundwater resources (Pham et al., 2019), of which the Mekong River basin experienced severe human interventions such as groundwater pumping, dams constructions, and city extension while the Amazon River basin is considered as one of the largest natural river basins with low urbanization and human activities (Xiong et al., 2022). It is discovered that the GCM simulations without bias correction show obvious underestimations over two regions with large uncertainty, however, which

have significantly reduced after bias correction along with a lower spread range (Figure R1). The amplitudes of the GCM series are adjusted to nearly the same as GRACE data, with the long-term trends unaffected.

Moreover, given the favorable consistency between CMIP6 GCMs TWS and both GRACE observations and in-situ measurements (Wu et al., 2021), our bias-correction based on GRACE data is expected to decrease their differences derived from the missing key TWSA components of the models by comparing with GRACE observations (Figures R2). Additionally, independent evaluation of GCM TWSA with and without bias-correction against the water balance estimates of TWSA changes from the observational products of CRU P, GLEAM ET, and GRUN R also presents a satisfactory correlation temporally and spatially (Figures R3-R4). The regions with good accuracy, like Alaska, western parts of the Tibetan Plateau, and northern Russia, decrease after bias correction. These differences over the high-latitude regions might be explained by the simplified treatment of permafrost in GCMs due to the prevailing uncertainties in, e.g., changes in thermo-physical properties of the soil during freezing and thawing cycles (Burke et al., 2020). On the contrary, the areas with relatively poorer accuracy before bias correction, such as North Africa and northern South America, slightly improve.

Notwithstanding the observed differences in some regions, our trend-preserving method used for bias correction would not influence the long-term trend estimations of both TWSA and TWS-DSI, and, therefore not impact our evaluation of the DDWW paradigm (Hempel et al., 2013). We take the ensemble mean of eight selected GCMs since all of them can simulate soil moisture and snow water. As suggested by the reviewer, we do not process the historical datasets similarly due to the different objects of different ensemble members (e.g., Table R1 and Table R2). It is noteworthy that the trend-preserving method would not affect the long-term trends of the GCM TWSA, and, therefore not influence our DDWW evaluation results in any way.

[Figure]

**Figure R1.** Monthly TWSA from GRACE and GCMs with and without bias correction in (a) Amazon and (b) Mekong River basins during 2002-2014. Note: The shading region means the spread of the GCM ensemble.

**Table R1.** Summary of attributes of different datasets used in this study.

| Dataset | GRACE | WGHM | VIC | Noah | CLSM | CMIP6 |
|---|---|---|---|---|---|---|
| Parameter | Satellite | GHM | | LSM | | GCM |
| Surface water storage | √ | √ | × | × | × | × |
| Soil moisture | √ | √ | √ | √ | √ | √ |
| Groundwater storage | √ | √ | × | × | √ | × |
| Canopy water | √ | √ | √ | √ | √ | × |
| Snow water | √ | √ | √ | √ | √ | √ |
| Soil layers (no.) | / | 1 | 3 | 4 | 10 | 5~10 |
| Soil depth (m) | / | 2 | 2 | 2 | 1 | 2~10 |

**Table R2.** Summary of the changes in the DDWW test results over global land during 1985-2014.

| Model/dataset | Previous results (ensemble mean of DATASET) | Updated results (individual datasets) [range] | Remark |
|---|---|---|---|
| DD | 16.7% | 6.47%-20.17% | From the perspective of TWSA, the DDWW is still challenged based on both the ensemble mean (previous version) and the individual datasets (current version) used in this study. |
| DW | 8.4% | 5.42%-16.13% | |
| WW | 11.4% | 4.54%-20.67% | |
| WD | 14.9% | 4.79%-19.3% | |
| TD | 2.1% | 0.95%3.88% | |
| TW | 1.8% | 0.73%-2.63% | |
| Non-significant | 45.1% | 17.2%-72.42% | |

[Figure]

**Figure R2.** (a) Probability density function and (b) Taylor diagram of NRMSE between TWSA derived from the GRACE mission and each member and the ensemble mean of eight GCMs during the period April 2002-December 2014. Solid and dashed lines in sub-figure (a) and corresponding filled circles and triangles in sub-figure (b) denote the original and bias-corrected time series.

[Figure]

**Figure R3.** Time series of the monthly changes in TWSA (TWSC) and water balance estimates (i.e., P-E-R) derived from GRACE, GCM, and observations during 2002-2014. Note: The shaded regions represent the spread of the CMIP6 ensemble.

[Figure]

**Figure R4.** Spatial distribution of correlation coefficient between monthly water balance estimates of TWSA changes and the ensemble mean of GCM data (a) before and (b) after bias corrections during 1985-2014. The blank grids indicate the missing values of the datasets.

References:

Burke, E.J., Zhang, Y., Krinner, G. 2020. Evaluating permafrost physics in the coupled model intercomparison project 6 (CMIP6) models and their sensitivity to climate change. Cryosphere., 14 (9) , pp. 3155-3174

Pham-Duc, B., Papa, F., Prigent, C., Aires, F., Biancamaria, S., and Frappart, F. 2019. Variations of surface and subsurface water storage in the Lower Mekong Basin (Vietnam and Cambodia) from multisatellite observations. Water, 11(1). https://doi.org/10.3390/w11010075

Hamed, K. H., Rao, A. R. 1998. A modified Mann-Kendall trend test for autocorrelated data. Journal of hydrology, 204(1-4), 182-196.

Hempel, S., Frieler, K., Warszawski, L., Schewe, J., Piontek, F., 2013. A trend-preserving bias correction: the ISI-MIP approach. Earth Syst. Dyn. 4, 219–236. https://doi.org/10.5194/esd-4-219-2013

Xiong, J., Yin, J., Guo, S., He, S., Chen, J, Abhishek. 2022. Annual runoff coefficient variation in a changing environment: a global perspective. 6, 064006. 10.1088/1748-9326/ac62ad.

Wu, R.-J., Lo, M.-H., Scanlon, B.R., 2021. The Annual Cycle of Terrestrial Water Storage Anomalies in CMIP6 Models Evaluated against GRACE Data. J. Clim. 34, 8205–8217. https://doi.org/10.1175/JCLI-D-21-0021.1

Zhao, M., Geruo, A., Velicogna, I., Kimball, J.S., 2017. Satellite Observations of Regional Drought Severity in the Continental United States Using GRACE-Based Terrestrial Water Storage Changes. J. Clim. 30, 6297–6308. https://doi.org/10.1175/JCLI-D-16-0458.1

Due accreditation and comparison of other approaches:

We regret the unintentional impression that we criticize or curtail the value (whether methods or applicability) of other approaches/metrics (e.g., P, P-E, etc.). We just intend to highlight the differences in the governing processes and hence the applicability of the various metrics. We have modified or rather weakened such instances in the revised manuscript. We also calculate a new metric called P-E-R as the residual of precipitation, evapotranspiration, and runoff, which represents the changes in TWSA in terms of the water balance equation (Famiglietti and Rodell, 2013). It means different from the TWSA and its derived metric (TWS-DSI), the latter represents the actual status of land over the long-term baseline, while the P-E-R means its changes. Thus, we have added the cross-comparison between the two indexes, attempting to investigate the mechanism of the variations in dryness/wetness of the land by comparing their differences and bridging the total mass changes to the land surface water balance. For example, a dry year in an agriculturally dominant basin (e.g., the Ganges basin in India) will trigger more groundwater extraction leading to a more acute decrease in TWSA (primarily due to evaporation losses) than the corresponding decline in P itself. In this case, the 'soil moisture' may exhibit positive trends, thus providing ambiguous interpretations. Such issues are not prevalent in our TWSA-based assessment. Therefore, although other indices (e.g., based on P, ET, soil moisture, or a combination thereof) may undoubtedly perform at par for the specific variable in question, they tend to present equivocal inferences for the total water storage. It can be easily understood by the example of soil moisture or evapotranspiration-based indices in a highly irrigated area such as the Ganges river basin. TWS is unremittingly declining due to the overexploitation of groundwater for agriculture in this region (Rodell et al., 2009), while E or soil moisture may have positive trends, thus attenuating the actual TWS situation. Such ambiguities across the prevailing metrics further strengthen our research hypothesis and objectives.

We have clarified our workflow in the method section. We hope our revisions will put forward our results in a more robust way.

References:

Famiglietti, J. S. Rodell, M. 2013. Water in the balance. Science. 340 (6138), 1300–1301. doi:10.1126/science.1236460.

Rodell, M., Velicogna, I., Famiglietti, J.S., 2009. Satellite-based estimates of groundwater depletion in India. Nature 460, 999–1002. https://doi.org/10.1038/nature08238

Results and mechanism: This study does not involve mechanism analysis, and does not analyze why some typical places are getting drier or wetter. Fig. 2 and Fig. 4 make no sense as they are another displays of the same results in Fig. 1 and Fig. 3. Although this division method was used in the IPCC6 and even considered popular by the authors, it did not bring any innovative insights to this study. Moreover, they are difficult to interpret. Instead, the readers are more care about how dryness/wetness changes in time and why there are changes happen.

Response: As suggested, we have added mechanism analysis based on the comparison with the new metric P-E-R and presented the temporal changes of dryness/wetness over the selected typical region of Qinghai-Tibetan-Plateau. Please also see the third subsection of the response above for further details. Moreover, we have removed Figures 2 and 4 in the updated manuscript and have restrained our analysis on the global land only. We have shared the data used in the manuscript figures, as well as the historical datasets and bias-corrected CMIP6 members to enable the reproducibility of the results at the required spatial scales (e.g., basin scales).

Innovation and significance: The authors need to rethink and justify what are the new results or developments reported in this study? Why are these new results or developments significant?

Response: We have highlighted the new findings and the significance reported by our study in the conclusion of the revised manuscript as follows:

Conclusion (Lines 583-622):

In this study, the historical TWS-DSI monthly time series over global land during 1985-2014 is calculated from an ensemble of two GHMs (VIC and WGHM), two LSMs (Noah and CLSM), and one GRACE reconstruction. In addition, future projections of TWS-DSI from 2071 to 2100 under SSP126, SSP245, and SSP585 scenarios are derived from the average of eight selected CMIP6 GCMs after bias-correction using GRACE observations. Subsequently, we detect the long-term trends in dryness/wetness in both the past and future periods based on TWS-DSI. Further, the DDWW paradigm has been evaluated with a significance level of 0.05 from the perspective of terrestrial water storage change. We also establish the metric P-E-R based on multiple observational products and from the same models as the TWS-DSI for comparison. The uncertainty sourced from different choices of models, methods, and confidence levels has been discussed systematically. The new findings are summarized as follows.

(1) During the historical period, the percentages of global land area presenting significant (p<0.05) drying and wetting trends range from 13.06% (WGHM)-43.35% (GRACE reconstruction) and 13.7% (CLSM)-39.43% (GRACE reconstruction),

respectively. The wetting trends are mainly in north Australia, north and South Africa, south and northwest China, western South America, central United States, and East Russia. While the drying trends are found in Arab region, west Brazil, Northeast Asia, and southern and northern American continent. During the future period under climate change, the proportion of drying areas (always ~10% higher than wetting) with a significant slope increases from SSP126 (19.52%) to SSP585 (29.04%) scenario. A similar change is detected in the percentage with significant wetting trends, which reaches 11.48%, 13.01%, and 18.42% under SSP126, SSP245, and SSP585 scenarios, respectively.

(2) A total of 11.01% (VIC) to 40.84% (GRACE reconstruction) of the global land area shows the DDWW paradigm valid, in which the drying and wetting area account for 6.47% (VIC)-20.17% (GRACE reconstruction) and 4.54% (VIC)-20.67% (GRACE reconstruction), respectively during the period 1985-2014. However, the area showing the opposite patterns, like "dry gets wetter" (DW) or "wet gets drier" (WD), account for the 10.21% (WGHM)-35.43% (GRACE reconstruction) of the global land, respectively. The proportion of areas supporting (opposing) the DDWW paradigm is 14.66% (16.76%), 14.26% (18.72%), and 17.08% (26.64%) under SSP126, SSP245, and SSP585 scenarios, respectively.

(3) Parallel estimates of the water balance variables and their comparison with the TWSA-based analysis, on the one hand, shed light on the governing mechanisms and translation of hydro-meteorological fluxes to the land water storage, on the other hand, outline additional insights into the varying and sometimes even contrasting behavior of the various metrics.

(4) Sensitivity analysis on different choices of significance levels from 0.01 to 0.1 for the long-term trends indicates similar patterns, in which the maximum decrease (increase) in the DDWW-validated regions reaches –7.4% (4.47% historically under the 0.01 (0.1) level, respectively. Such consistency is also evidenced by the projected TWS-DSI in the future under various scenarios. Moreover, independent experiments based on the individual TWSA datasets suggest that the divergent data sources might lead to model-variable biases for both the DDWW-agreed and DDWW-opposed patterns. The use of distinctive GCMs also suggests slightly overrated (e.g., GFDL-ESM4) and underrated (e.g., CanESM5) percentages of such patterns in the future under multiple emission scenarios.

New insights from the TWSA perspective highlight that the widely-used DDWW paradigm is still challenged in both historical and future periods under climate change. The differences between test results based on P-E-R imply the robustness of our developed TWS-DSI in capturing the total land water variations induced by climate changes and human activities, suggesting potentially new knowledge in the land hydrology field. The regional aggregation of our study in the Qinghai-Tibetan Plateau can provide important inferences for decision-makers and stakeholders for the sustainable management and efficient utilization of water resources under global change.

**Specific comments:**

(1) Line 9-10 and Line 17-18: These statements are contradicted. You are saying the DDWW is challenged due to the choice of different metrics and datasets used, but you also stated the different data sources have subtle influences on the evaluation results.

Response: We have modified the introductory statement in the revised manuscript as follows (Lines 8-9):
However, the paradigm is largely conditioned by the choice of different metrics and datasets used and is still unexplored from the perspective of terrestrial water storage anomaly (TWSA).

(2) Line 21: "The hydrological conditions of the land surface have experienced...". The first sentence of this manuscript is talking about land surface condition. This is contrary to the author's argument that they are not concerned with the surface dryness/wetness, but with the entire land system.

Response: We regret the misleading articulation. We have modified the text, which now follows the order: Introduction and importance of hydrological cycle>DDWW paradigm>literature review dealing with P-E>introducing TWS>research hypothesis and objectives of our study.

(3) Line 37: What are oceanic records?

Response: It means the oceanic observations that provide environmental information for marine management, such as air temperature, precipitation, and evaporation (OOPC, Ioc-goos-oopc.org. Retrieved 11 June 2022). We have revised it.

(4) Line 45: P-ET is the amount of water remaining in the land system, but the components in the GCMs (soil moisture and snow water) and VIC (moisture), Noah (soil moisture, snow, and canopy water) are parts of the water stored in the land system (Table S2), and thus the models lack some components of the terrestrial water storage.

Response: We understand the reviewer's point of view. However, we would like to take this opportunity to explain the difference that hinges on 'hydroclimate' variables and the overall status of the land water storage, i.e., 'TWS'. Although absolute hydroclimatic variables and their changes are interrelated by the conservation of water mass and energy, their magnitude of change may not be consistent (Roth et al., 2021; Huntington et al., 2006; Dirmeyer et al., 2016; Labat et al., 2004). For example, an increase in precipitation in time does not necessarily imply an increase in river water availability—if accompanied by a steep increase in evaporation by more thermal energy availability, runoff can, in fact, decrease (Bosson et al., 2012, Katul et al., 2022).

Therefore, based on changes in precipitation or evapotranspiration, or runoff alone, it cannot be concluded how will be the variability of the total water storage.

Since all these hydroclimate variables are intricately affected by natural or human factors or a combination thereof, an out-and-out separation of these two convoluted factors is almost not possible. This becomes acute in the regions of dominant human activities. For example, a dry year in an agriculturally dominant basin (e.g., the Ganges basin in India) will trigger more groundwater extraction leading to a more acute decrease in TWSA than the corresponding decline in P itself.

Moreover, as we discuss above, the residual of precipitation, evapotranspiration, and runoff could be considered as changes in TWSA (TWSC) in terms of the water balance equation, instead of the TWSA itself. In other words, TWSA represents the current status of the land system over the long-term average, while TWSC denotes its change. Given the divergent meaning of the two variables, we conduct cross-comparisons between TWS-DSI and P-E-R, and analyze the mechanisms involved by comparing their differences. Lastly, we have explicitly discussed the inevitable uncertainties arising from various climate forcing, inadequate model physics, or the simplified assumptions of various processes.

References:
Roth, N., Jaramillo, F., Wang-Erlandsson, L., Zamora, D., Palomino-Ángel, S., Cousins, S. A. 2021. A call for consistency with the terms 'wetter'and 'drier'in climate change studies. Environmental Evidence, 10(1), 1-7.
Huntington TG. 2006. Evidence for intensification of the global water cycle: review and synthesis. J Hydrol. 319(1):83–95.
Dirmeyer PA, Yu L, Amini S, Crowell AD, Elders A, Wu J. 2016. Projections of the shifting envelope of water cycle variability. Clim Change. 136(3):587–600.
Labat D, Goddéris Y, Probst JL, Guyot JL. 2004. Evidence for global runoff increase related to climate warming. Adv Water Resour. 27(6):631–42.
Bosson E, Sabel U, Gustafsson LG, Sassner M, Destouni G. 2012. Influences of shifts in climate, landscape, and permafrost on terrestrial hydrology. J Geophys Res Atmos. 117(5):1–12.
Katul GG, Oren R, Manzoni S, Higgins C, Parlange MB. 2012. Evapotranspiration: a process driving mass transport and energy exchange in the soil-plant-atmosphere-climate system. Rev Geophys. 50(3):RG3002.

(5) Line 70: Is long-term P-ET approximately equal to the change in terrestrial water storage (TWS)? Why the authors do not use P-ET to construct an index and to perform the prediction of TWSA? Instead, this study uses partial outputs of soil moisture/snow data in the GCMs.

Response: As per the water balance equation, P-ET equals R+TWSC, i.e., summation of runoff and change in TWSA between two subsequent months (i.e., $TWSA_i$-$TWSA_{i-1}$). To be consistent in the variables (TWSA and TWSC) and to account for the water balance closures (as suggested by Reviewer#2), we used the metric P-E-R (=TWSC). However, as we discuss in the previous responses, they have different meanings for the terrestrial water cycle. Generally, it is difficult to detect significant trends in TWSC due to the slight inter annual variability at the yearly time scale (Lv et al., 2021). Thus comparing their differences, as done in the revised manuscript, appears better than the individual investigation based on either of them.

Reference

Lv, M., Ma, Z., Yuan, N. 2021. Attributing terrestrial water storage variations across China to changes in groundwater and human water use. Journal of Hydrometeorology, 22, 3– 21. https://doi.org/10.1175/jhm-d-20-0095.1

(6) Table 1: What are the differences between GRACE reconstructions and GRACE mascons solutions?

Response: The GRACE mascon solution is a type of GRACE-observed TWSA solution that has been widely used in the hydrology community (Scanlon et al., 2018). GRACE reconstruction is an ML-based TWSA product based on the GRACE observations and multi-source meteorological inputs, which was proposed to overcome the relatively short time span of the GRACE and GRACE-Follow On missions (~20 years). We have added more descriptions for the GRACE solutions and GRACE reconstructions in the revised manuscript as follows (Lines 132-142):

The GRACE (and GRACE Follow-On) missions have provided unprecedented estimates of monthly TWSA worldwide from April 2002 up to the present, however, with the 33 months missing because of the instrumental issues and mission interruption (Tapley et al., 2004). We use the GRACE mascon solution from the Center for Space Research at the University of Texas at Austin (UTCSR) to serve as the benchmarking product from the period 2002-2014 (Watkins et al., 2015). Compared to conventional GRACE products (e.g., spherical harmonic solutions), mascon solutions do not need spatial (e.g., smoothing) or spectral (e.g., de-striping) filtering or other empirical scaling and therefore have higher signal-to-noise ratio, higher spatial resolutions, and eventually reduced errors (Save et al., 2016; Watkins et al., 2015). However, the GRACE observational products were not adequate to assess the long-term trends of TWSA due to relatively short temporal coverage (~20 years). Therefore, we obtain the GRACE reconstruction provided by Li et al. (2021b) for evaluation of the DDWW paradigm, which is generated using state-of-the-art machine learning and statistical methods and is also trained by the consistent GRACE mascon product from the UTCSR institution.

Reference:
Scanlon, B.R., Zhang, Z., Save, H., Wiese, D.N., Landerer, F.W., Long, D. Longuevergne, L., Chen. J. 2016. Global evaluation of new GRACE mascon products for hydrologic applications. Water Resour. Res., 52 (12), pp. 9412-9429

(7) Figure 2 makes no sense and it is hard to interpret. Instead, the manuscript can present dryness/wetness changes in some key regions here.

Response: We have removed Figure 2 in the revised manuscript and have restrained our analysis on the global land only. Since we have provided the processed data publicly available, further studies may focus on different regions (e.g., SREX regions, basin scales) as per the question in focus. Moreover, we have provided the temporal changes over the Qinghai-Tibetan Plateau, and the global land in the revised version to better establish the mechanism analysis as suggested by the reviewer.

(8) Line 273-276: Why did the authors use AI derived from CRU data to define wet and dry zones rather than TWS-DSI? The following DDWW analyses are based on the changes of TWS-DSI.

Response: Because the TWS-DSI, as a normalized drought index, is zero when looking at the long-term average and therefore not appropriate for the static classification for the climate zones, though we define the increase/decrease in TWS-DSI as wetting/drying signals. TWSA broadly shares a similar variability since it represents the anomaly over the long-term baseline. Therefore, in this case, we use commonly used AI to classify regions as arid, humid, or transitional following previous studies encountering similar issues (e.g., Feng and Zhang, 2015, Hu et al., 2019, and Yang et al., 2019). We have clarified the reasons for using AI for climate classification in the methods section. In addition, we also compare our AI-based results with the widely used Köppen-Geiger climate classification maps in the revised manuscript as follows (Lines 226-239):

To evaluate the DDWW paradigm over global land, the effective Aridity index (AI) is used to classify a grid cell as an arid, humid, and transitional region following Yang et al. (2019) because TWS-DSI/TWSA approximates zero for the long-term mean. The AI is calculated as the ratio of annual precipitation to potential evapotranspiration provided by the CRU TS-v4.06 during the same period as TWS-DSI (i.e., 1985-2014). The global distribution of multi-year average AI and the classifications during the period 1985-2014 is presented in Figure S3, which is also highly consistent with the widely used Köppen-Geiger climate classification maps (Beck et al. 2018) (Figure S2). It can be seen that most of the arid regions (AI<0.5) are located in southwestern America, north and south Africa, central Asia, Arabian regions, and Australia, accounting for 39.3% of the land. The percentage of humid areas (AI>0.65) that are mainly located in east America, the Amazon region, central Africa, south China, west Europe, and Russia reaches 52.8% of the land. An approximate 7.9% of the land area is defined as the transitional region, referring to an intermediate between arid and humid climates. The transitional region generally lies in the shared boundaries of the humid and arid regions (e.g., western America, northern Canada, central Asia, western Africa, East Russia, and Australia). The DDWW paradigm is evaluated at a 5% significance level in this study, combined with the standard AI-derived climate classifications. We calculate the global mean trends of TWS-DSI using a spatially weighted method to account for the changing area of grid cells with latitudes.

[Figure]

**Figure S2.** Global distribution of the improved Köppen-Geiger classifications during the period 1980-2016. Note: Please refer to Beck et al. (2018) for the details of the classification criteria. The dashed boundary represents the Qinghai-Tibetan Plateau.

[Figure]

**Figure S3.** Global distribution of the (a) multi-year average aridity index (AI) and (b) climate type during the period 1985-2014. Note: The regions where AI>0.65 and <0.50 are defined as humid and arid regions, respectively.

References:

Beck, H.E., Zimmermann, N.E., McVicar, T.R., Vergopolan, N., Berg, A., Wood, E.F. 2018. Present and future Köppen-Geiger climate classification maps at 1-km resolution. Sci. Data, 5, https://doi.org/10.1038/sdata.2018.214

Feng, H., Zhang, M., 2015. Global land moisture trends: drier in dry and wetter in wet over land. Sci. Rep. 5, 18018. https://doi.org/10.1038/srep18018

Hu, Z.Y., Chen, X., Chen, D.L. Li, J.F., Wang, S., Zhou, Q., Yin, G., Guo, M. 2019. "Dry gets drier, wet gets wetter": a case study over the arid regions of central Asia. Int J Climatol 39(2):1072–1091

Yang, T., Ding, J., Liu, D., Wang, X., Wang, T., 2019. Combined Use of Multiple Drought Indices for Global Assessment of Dry Gets Drier and Wet Gets Wetter Paradigm. J. Clim. 32, 737–748. https://doi.org/10.1175/JCLI-D-18-0261.1

(9) Line 281: "We compare AI and TWSA derived from DATASET and CMIP6 between 1985 and 2014 in Figure S6". Is it a result comparison between Figure S5 and Figure S6 here?

Response: Yes, it is a result comparison between Figures S5 and S6. However, we have removed this because we do not simply take the ensemble mean of the DATASET as suggested by the reviewer. Moreover, we have added a comparison between AI and the Köppen-Geiger climate classification maps in the new version, as discussed above.

(10) Line 273-283: These contents about how to operate should be adjusted to the method section?

Response: As suggested, we have moved these contents to the method section in the revised version.

(11) Line 284: "Figure 3 illustrates the test results...". This is not a good way to express the content of figures.

Response: We have revised this statement in the new version as follows (Lines 429-430):
Combined with the climate regions classified by AI, we further test the DDWW paradigm at a 5% significance level using both TWS-DSI and P-E-R over global land in the past and future (Figures 3 and 4).

(12) Line 324-325: "Greve and Senevirtne (2015) used climate projections from CMIP5 to establish the measure for assessment of the DDWW paradigm...". The method used by Greve (2015) is more acceptable to peers.

Response: As suggested, we have established the new metric P-E-R for comparison with the TWS-DSI in the revised manuscript and have thoroughly discussed the observed similarities and differences.

(13) The presentation of Figure 4 makes no sense and it is hard to interpret. It may be more interesting to modify it to temporal changes over key areas.

Response: We have removed Figure 4 in the revised manuscript and have restrained our analysis on the global land only. Since we have provided the processed data publicly available, further studies may focus on different regions (e.g., SREX regions, basin scales). Moreover, we have provided the temporal changes over the selected key area of Qinghai-Tibetan-Plateau and the global land in the revised version as suggested by the reviewer.

(14) Line 355-358: "...resulting in the lack of certain TWSA components.". Why the authors do not use P-ET derived from the models to represent TWSA? In this case, none of the flaws discussed here exist.

Response: As discussed above, we have additionally established the metric P-E-R for comparison with TWS-DSI. However, it is noteworthy that the counterpart of precipitation, evapotranspiration, and runoff is changes in TWSA (i.e., TWSC) rather than TWSA in terms of the water balance equation. They reflect different aspects of the water cycle and can help to reveal the mechanisms involved by highlighting the difference between these metrics. To this end, we reserve the evaluation based on TWS-DSI and link these differences with some suggestions for future research in the discussion section of the revised manuscript.

(15) Line 417-419: Please explain what are the advantages of the developed TWS-DSI?

Response: We have added the advantages of the developed TWS-DSI in the revised manuscript as follows (Lines 208-213):
The non-dimensional TWS drought severity index (TWS-DSI) is established at both $1° \times 1°$ grid cell and regional/global scales, which is normalized by the regional hydro-climatological variability because a given magnitude of TWS deficit could indicate different dryness/wetness conditions in different climate regions. TWS-DSI has clear classification categories based on U.S. Drought Monitor (USDM) and is suitable for comparing dryness/wetness status for different locations and periods (Table S2). It has been widely used in hydrology and climate fields due to its simple structure and effective ability to capture drying and wetting conditions (Pokhrel et al., 2021).

(16) Line 447-462: The authors need to refine the conclusions. These do not look like conclusions, at least not serving for the purpose of this study.

Response: We have added some key points to the conclusion and refined the structure as suggested by the reviewer. Please also see our response to the major comment "Innovation and significance".

(17) The authors need to recheck and simplify the expression and logic of the entire manuscript, as many expressions seem redundant and use uncommon words, making it difficult to read.

Response: While revising the manuscript according to the reviewer's comments, we have thoroughly checked, simplified the expressions, and proofread the text to improve its readability.

**Legend**

Reviewers' comments

Authors' responses

Direct quotes from the revised manuscript

We thank the reviewer for his time in reading our manuscript and very detailed comments on our manuscript. Point-by-point replies to the comments or suggestions made can be found below. Overall, we have made the following major changes to the manuscript:

- Performed additional analysis using P-E-R and analyzed and compared the results with TWSA-DSI.
- Instead of showing only the ensemble mean of various model and observation-based results, we have now shown the results from individual datasets during the historical period (1985-2014).
- Added extensive discussion about the various mechanisms and governing processes for the observed patterns and the similarities and disparities from the previous studies.

**Reviewer #2:** The manuscript examines the dry gets drier and wet gets wetter (DDWW) paradigm from a water storage perspective over the terrestrial fraction of the water cycle. The topic is contemplated within the scope of Hydrology and Earth System Sciences via the study of the spatial and temporal characteristics of global water resources and would be of interest to its readership. It is a topic that remains under debate in the last decade and studies that bring new evidence, such as this, can substantially impact the ongoing research. The analysis based on the terrestrial water storage drought severity index is reasonable and aims to quantify wet/dry regime trends under different warming scenarios and assess the agreement or rebuttal of the DDWW paradigm. The study has clear research hypothesis and objectives, that are reflected in the results, and help us increase our understanding of the changes in the global water cycle. However, there are some important aspects that should be addressed before being considered for publication.

Response: We thank the reviewer for his highly encouraging feedback on the manuscript. All the concerns raised have been addressed in the revised manuscript. We hope the modified text, along with the supplementary analyses and discussions, will put forward the results in a much more robust way.

Before moving to the specific revision comments, it is important to note the lack of consistency across the hydrologic and climatic communities in the use of the terms "wet/wetter" and "dry/drier", which in some cases can be misleading (Roth et al. 2021). Thus, it is easy to misinterpret whether a variable is truly appropriate for

describing wetting or drying over a region. In this study, a variable (terrestrial water storage) which is not directly involved in the formulation of DDWW paradigm (precipitation/evaporation) is used to validate the paradigm itself. This raises concerns about the applicability of terrestrial water storage as a metric that can confirm or falsify the DDWW hypothesis. The study needs to convincingly prove this. A feasible way to achieve it would be to compare the current findings with P – E, taken from the models (GHMs, LHMs, and CMIP6). This also holds the opportunity to highlight the mechanisms involved in the observed changes and/or pinpoint the biases in the models. Some caution should be taken here since the comparison should also consider surface runoff to satisfy the budget closure. In any case, though, this can help to bridge the different methodologies and explore their complementarity.

Response: We thank the reviewer for cautioning about the interpretation of the topical issue of DDWW and for guiding us through the additional assessment based on the water balance-based metric (i.e., P-E-R). Please find below the parsed-out details about the changes made during the revision of the manuscript.

Inferences of the "wet/wetter" and "dry/drier" terms: We fully agree that the various environmental disciplines define 'wetter' and 'drier' differently due to the different temporal scales (e.g., from <1 year to >20 years) and spatial extents (i.e., from local to a global extent) within the multidisciplinary climate change communities (Roth et al. 2021). However, the general concept of the terms "wetter"/"dries" imply the increase/decrease of the available amount of freshwater over a specific region. In this case, the TWSA (and TWSA-derived index), constituting the total water mass in the land system, is theoretically reasonable to represent the "wetter" and "drier" global land (Yi et al., 2016; Long et al., 2018). Moreover, we have clarified the meaning, spatial and temporal scales, and the original state that the "wetter" and "drier" were compared to in the methods section as follows (Lines 207-213):

TWSA, consisting of the water volume stored in the land surface and subsurface, is applied to define the "wetting" and "drying" conditions of the landmass in this study. The non-dimensional TWS drought severity index (TWS-DSI) is established at both 1°×1° grid cell and regional/global scales, which is normalized by the regional hydro-climatological variability because a given magnitude of TWS deficit could indicate different dryness/wetness conditions in different climate regions. TWS-DSI has clear classification categories based on U.S. Drought Monitor (USDM) and is suitable for comparing dryness/wetness status for different locations and periods (Table S2). It has been widely used in hydrology and climate fields due to its simple structure and effective ability to capture drying and wetting conditions (Pokhrel et al., 2021).

Need for and applicability of TWSA as a metric for examining DDWW: Although the DDWW paradigm was initially formulated using the metric "P-ET" (Held and Soden, 2006; Greve et al., 2014), it is worth noting that independent examinations from multiple aspects, for example, soil moisture (Feng and Zhang, 2015) and runoff (Yang

et al., 2019), are attracting increasing attention in the last decade. Therefore, the evaluation of the DDWW paradigm from the TWSA perspective can potentially provide new evidence for the community working on, e.g., ecosystem functioning (Humphrey et al., 2018), sea-level rise (Jeon et al., 2018), water budget (Sheffield et al., 2009), and freshwater availability (Rodell et al., 2018). Moreover, TWSA has not been widely applied as a "wetter" and/or "drier" metric due to the lack of observations globally till the launch of GRACE in 2002. To this end, we have revised the title of the manuscript to prevent using the term "re-examine".

Additional analysis and comparison with 'P-E-R' metric: As suggested by the reviewer, we have additionally established the land water balance metric P-E-R for comparisons with TWS-DSI to highlight the difference between these metrics and to discuss the mechanisms involved. All three constituent variables, i.e., P, ET and R, were taken from the same models (i.e., GHMs, LSMs, and GCMs) as those for TWS-DSI to account for the uncertainty associated with the models and meteorological forcing data. We also used an observation-based combination of P-E-R using precipitation from CRU TS, evapotranspiration from GLEAM, and runoff from GRUN datasets. The inter-comparison between TWS-DSI and P-E-R can help us to bridge the different methodologies and reveal the mechanisms and bias in the changes in dryness/wetness.

References:

Feng, H., Zhang, M., 2015. Global land moisture trends: drier in dry and wetter in wet over land. Sci. Rep. 5, 18018. https://doi.org/10.1038/srep18018

Greve, P., Orlowsky, B., Mueller, B., Sheffield, J., Reichstein, M., Seneviratne, S.I., 2014. Global assessment of trends in wetting and drying over land. Nat. Geosci. 7, 716–721. https://doi.org/10.1038/NGEO2247
Humphrey, V., Zscheischler, J., Ciais, P. et al. Sensitivity of atmospheric CO2 growth rate to observed changes in terrestrial water storage. Nature 560, 628–631 (2018). https://doi.org/10.1038/s41586-018-0424-4

Jeon, T., Seo, K-W., Youm, K., Chen, J., Wilson, C.R. 2018. Global sea level change signatures observed by GRACE satellite gravimetry. Scientific Reports. 8(1): 13519. http://dx.doi.org/10.1038/s41598-018-31972-8.

Long, B., B. Q. Zhang, C. S. He, R. Shao, and W. Tian, 2018: Is there a change from a warm-dry to a warm-wet climate in the Inland River area of China? Interpretation and analysis through surface water balance J. Geophys. Res.: Atmos., 123, 7114–7131, https://doi.org/10.1029/2018jd028436.

Roth, N., Jaramillo, F., Wang-Erlandsson, L., Zamora, D., Palomino-Ángel, S., & Cousins, S. A. (2021). A call for consistency with the terms 'wetter'and 'drier' in climate change studies. Environmental Evidence, 10(1), 1-7.

Sheffield, J., Ferguson, C. R., Troy, T. J., Wood, E. F. and McCabe, M. F. 2009. Closing the terrestrial water budget from satellite remote sensing Geophys. Res. Lett. 36. L07403.

Yang, T., Ding, J., Liu, D., Wang, X., Wang, T., 2019. Combined Use of Multiple Drought Indices for Global Assessment of Dry Gets Drier and Wet Gets Wetter Paradigm. J. Clim. 32, 737–748. https://doi.org/10.1175/JCLI-D-18-0261.1

Yi, S.; Sun, W.K.; Feng, W.; Chen, J.L. Anthropogenic and climate-deiven water depletion in Asia. Geophys. Res. Lett. 2016, 43, 9061–9069.

Rodell, M., Famiglietti, J.S., Wiese, D.N. et al. Emerging trends in global freshwater availability. Nature 557, 651–659 (2018). https://doi.org/10.1038/s41586-018-0123-1

**Specific Comments**

(1) Lines 1-2: The term "re-examination" should be reconsidered and perhaps be replaced with something like "alternative/complementary examination".

Response: Thank you for the suggestion. After carefully considering the comments from Reviewers #1 and #2, we consider it best to drop the word 're-examination' from the title and update the title to - 'Global evaluation of the dry gets drier and wet gets wetter paradigm from terrestrial water storage changes perspective'.

(2) Lines 25-27: It would be more appropriate to reference the original study of Held and Soden (2006).

Response: Thank you for the suggestion. We have referenced the original study.

(3) Lines 34-42: You might want to look at the work of Roderick et al. (2014).

Response: We have read through the suggested key study and have included relevant information in our manuscript.

(4) Lines 43-74: This paragraph could be split into two (line 56 perhaps?) to improve readability.

Response: We have split the paragraph as suggested.

(5) Line 84 (Table 1): Since the three GRACE reconstructions come from two papers it could help the readers if they were also somehow distinguished in the table.

Response: Only the GRACE (CSR) reconstruction from Li et al. (2021) is used for the evaluation of the DDWW paradigm in the new version owing to high reliability and robustness (produced using a combination of three data-driven approaches and compared against various hydroclimatic indices and water storage component outputs by global hydrological models) of this product compared to the other ones (Li et 2020; 2021). We have modified the table accordingly.

References:
Li, F., Kusche, J., Chao, N., Wang, Z., Loecher, A., 2021. Long-Term (1979-Present) Total Water Storage Anomalies Over the Global Land Derived by Reconstructing GRACE Data. Geophys. Res. Lett. 48, e2021GL093492. https://doi.org/10.1029/2021GL093492
Li, F., Kusche, J., Rietbroek, R., Wang, Z., Forootan, E., Schulze, K., Lück, C. 2020. Comparison of Data-driven Techniques to Reconstruct (1992-2002) and Predict (2017-2018) GRACE-like Gridded Total Water Storage Changes using Climate Inputs[J]. Water Resources Research, 56(5), e2019WR026551. https://doi.org/10.1029/2019wr026551.

(6) Lines 84-85: Averaging the datasets always comes with certain challenges. For example, in this case we would expect that the three reconstructions of GRACE are strongly correlated and thus their impact to the estimation of the mean would be stronger. A cross-correlation matrix between the datasets would help to assess the magnitude of the impact and comment on it in the manuscript.

Response: Since different models/products have different TWSA definitions, we opt to demonstrate the individual examination results of different datasets rather than merely the ensemble mean. The additional cross-correlation matrix with the GRACE observations shows the reasonable consistency of the datasets with the Pearson correlation coefficient ranging from 0.79 (Noah) to 0.99 (GRACE reconstruction). We have added this information in the revised manuscript.

[Figure]

**Figure R1.** Correlation matrix between the DATASET members and GRACE over global land excluding Antarctica and Greenland during April 2002-December 2014. Note: The numbers mean the Pearson correlation coefficients within two variables.

(7) Lines 87-88: Can you please provide some information about the percentage of the missing values, as well as their distribution among the number of consecutive missing values?

Response: We have provided information about the percentage and distribution of the missing months in the revised manuscript, as below (Lines 105-107).
The missing months (12% of the months, i.e., June 2002, July 2002, June 2003, January 2011, June 2011, May 2012, October 2012, March 2013, August 2013, September 2013, February 2014, July 2014, December 2014) of GRACE measurements have been filled using a linear interpolation method.

Response: We have removed "kinds" from both the places as suggested.

(9) Lines 167-168: Please also cite the original paper of Hempel et al. (2013) which has been used by Xiong et al. (2022).

Response: We have also cited the primary reference as suggested.

(10) Lines 177-178: Before applying linear interpolation and assessing the statistical significance of the slope the auto-correlation structure of the time series should be investigated. High values of auto-correlation coefficient could result to biased estimates of t, so if this is the case, alternative methods of slope significance should be applied (Hamed and Rao, 1998; Yue et al. 2002). In addition, the reference to the work of Greve et al. (2014) should be revisited as a different statistical test is applied at that study than the t-test.

Response: We have investigated the first-order autocorrelation structure of the time series of multiple datasets and the CMIP6 GCMs using the Durbin-Watson test (Figure R2) (Durbin and Watson, 1950, 1951). A total of 20% (GRACE reconstruction), 43% (WGHM), 41% (VIC), 23% (CLSM), 29% (Noah), and 20% (GCM) of the grid cells do not present autocorrelation during the historical period 1985-2014. For the future period, the percentage is 25%, 26%, and 22% under the SSP126, SSP245, and SSP585 scenarios, respectively. In this case, we select the modified Mann-Kendall trend test to obtain the true variance under the autocorrelation structure displayed (Hamed and Rao, 1998). In addition, we have changed the reference to the work of Greve et al. (2014) in the revised manuscript.

[Figure]

**Figure R2.** Global assessment of the autocorrelation during the (a-f) historical (1985-2014) and future (2071-2100) period under (b) SSP126, (c) SSP245, and (d) SSP585 scenarios. Note: The historical results are based on the (a) GRACE reconstruction, (b) WGHM, (c) VIC, (d) CLSM, (e) Noah, and (f) ensemble mean of eight GCMs, respectively using the Durbin-Watson test. The future results are based on the ensemble of eight GCMs. Generally, the residuals are considered not correlated when the Durbin-Watson test statistic has a value between 1.5 and 2.5. If the statistic is below 1 or above 3, then there is definitely autocorrelation among the residuals.

Reference:
Durbin, J., Watson, G. S. 1950. Testing for Serial Correlation in Least Squares Regression, I. Biometrika 37 : 409-428.

Durbin, J., Watson, G. S. 1951. Testing for Serial Correlation in Least Squares Regression, II. Biometrika 38: 159-179.

Hamed, K. H., Rao, A. R. 1998. A modified Mann-Kendall trend test for autocorrelated data. Journal of hydrology, 204(1-4), 182-196.

(11) Line 180: Should the region be considered as "uncertain" or should it be considered a region with no or non-significant long-term change?

Response: We have changed the terminology from "uncertain" to the "non-significant" regions throughout the revised manuscript.

(12) Lines 183-184: It is preferable to keep a single tense for the whole manuscript (past/present). This comment also applies for other lines.

Response: We regret the non-uniformity. We have kept the present tense for the whole of the revised manuscript.

(13) Line 185: Citation of the report is missing.

Response: Since we have excluded the case analysis based on the IPCC AR6 SREX regions in the revised manuscript according to the suggestions from Reviewer #1, the citation of the report is not needed anymore.

(14) Lines 190-208: My understanding after reading the bias-correction method of Xiong et al. (2022) is that the CMIP6 ensemble was bias-corrected using GRACE TWSA. If this holds true, then the evaluation of the TWSA derived from the ensemble mean of CMIP6 raises some questions about its validity and therefore NRMSE is lower compared to DATASET.

Response: In addition to the comparison to the GRACE TWSA, we also provided the evaluation of the bias-corrected TWSA changes against the water balance estimates (i.e., P-E-R) during 1985-2014 (Figures R3 and R4). The observation-based water balance estimates correlate well with GRACE TWSA and GCM-modelled P-E-R with a correlation coefficient of 0.62 and 0.93, respectively. The GCM-simulated changes in TWSA also present a strong correlation with the observational products before and after bias correction. The spatial distribution of correlation coefficients between TWSC from observations and GCMs with and without bias correction shows the performances in regions with good accuracy, like Alaska, western parts of the Tibetan Plateau, and northern Russia, decrease after bias correction, which might be caused by the simplified treatment of permafrost in GCMs due to the prevailing uncertainties in, e.g., changes in thermo-physical properties of the soil during freezing and thawing cycles (Burke et al., 2020). On the contrary, the areas with relatively poorer accuracy before bias correction, such as North Africa and northern South America, slightly improve after bias correction. Notwithstanding the observed differences in some regions, our trend-preserving method used for bias correction would not influence the long-term trend estimations of both TWSA and TWS-DSI and therefore does not

impact our evaluation of the DDWW paradigm (Hempel et al., 2013).

[Figure]

**Figure R3.** Time series of the monthly changes in TWSA (TWSC) and water balance estimates (i.e., P-E-R) derived from GRACE, GCM, and observations during 2002-2014. Note: The shaded regions represent the spread of the CMIP6 ensemble.

[Figure]

**Figure R4.** Spatial distribution of correlation coefficient between monthly water balance estimates of TWSA changes and the ensemble mean of GCM data (a) before and (b) after bias corrections during 1985-2014. The blank grids indicate the missing values of the datasets.

References:

Burke, E.J., Zhang, Y., Krinner, G. 2020. Evaluating permafrost physics in the coupled model intercomparison project 6 (CMIP6) models and their sensitivity to climate change. Cryosphere., 14 (9) , pp. 3155-3174

Hempel, S., Frieler, K., Warszawski, L., Schewe, J., Piontek, F. 2013. A trend preserving bias correction–the ISI-MIP approach. Earth System Dynamics, 4(2), 219-236.

(15) Lines 214-216: Is there any likely explanation about the increase in the range of DATASET after 2010? Perhaps it can be linked to the decline of TWSA of a specific dataset.

Response: The reviewer is correct in his anticipation. This is caused by the abrupt changes in TWSA from the PCR-GLOBWB model. In the revised manuscript, since 1) we have excluded the usage of this model (because we attempt to construct the metric P-E-R for parallel comparison and it is not available for the PCR-GLOBWB model) and 2) opt to present the individual members of different datasets instead of just the ensemble mean, the increase in the range of DATASET after 2010 has disappeared (Figure R5).

[Figure]

**Figure R5.** Time series of monthly TWSA derived from the GRACE products and different TWSA datasets over the global land excluding Antarctica and Greenland during the period April 2002-December 2014. Note: NRMSE between GRACE and different datasets are also shown. The deep blue line denotes the ensemble mean of eight GCMs. The shaded areas represent the range of TWSA values among the individual GCM datasets.

(16) Line 219: It would be helpful to remind the readers that for the historical period DATASET is used, and not only the CMIP6 data that are used for the scenarios.

Response: We have clarified the data source for the trend estimate for the historical and future periods in both text and figure captions again.

(17) Line 221: Since you are referring to the slope "/a" is redundant. Still if you would like to keep it, you could consider replacing it with "/yr".

Response: We have replaced it with "/yr" throughout the revised manuscript.

(18) Line 228: I think "increasing" is the correct word here.

Response: Thank you. We have updated as suggested.

(19) Lines 228-229: There is no study related to S. Europe in the references. Most importantly, in Figure 1b it appears that SE. Europe is wet and the rest of the south not significantly drier. This contradicts older studies reporting a drying trend over the Mediterranean (e.g., Hoerling et al. 2012) and could shed new light to the ongoing discussion about the current and future conditions of S. Europe, so I would recommend elaborating more.

Response: As indicated by the reviewer, the reduction in winter precipitation has become a regular phenomenon in the Mediterranean and caused increased seasonal drought risks (Hoerling et al. 2012, Wagner et al., 2019). However, no statistically significant long-term decreasing trends are detected at the annual timescale for the region (Peña-Angulo et al., 2020; Vicente-Serrano et al., 2020). In particular, the centre of the Mediterranean basin including France, Italy, Croatia, and Slovenia presented insignificant decreasing trends in annual precipitation, while the western and eastern regions such as Spain and Greece demonstrated increasing trends during 1991-2018 (Caloiero et al., 2018; Peña-Angulo et al., 2020). These findings are generally consistent with our results based on TWS-DSI derived from different models during the historical period 1985-2014 (Figures R6 and R7).

We have revised this statement and added references related to the drying/wetting trends over South Europe. Moreover, we have added discussions for the past and future changes in wetness/dryness over southwestern Europe in the revised version.

[Figure]

**Figure R6.** Global distribution of the significant (p<0.05) long-term trends in TWS-DSI during (a-f) the historical (1985-2014) and future (2071-2100) period under (g) SSP126, (h) SSP245, and (i) SSP585 scenarios. Note: The historical results are based on the (a) GRACE reconstruction, (b) WGHM, (c) VIC, (d) CLSM, (e) Noah, and (f) ensemble mean of eight GCMs, respectively. The future results are based on the ensemble of eight GCMs.

[Figure]

**Figure R7.** Global distribution of the classification in long-term trends in TWS-DSI during (a-f) the historical (1985-2014) and future (2071-2100) period under (g) SSP126, (h) SSP245, and (i) SSP585 scenarios. Note: The historical results are based on the (a) GRACE reconstruction, (b) WGHM, (c) VIC, (d) CLSM, (e) Noah, and (f) ensemble mean of eight GCMs, respectively. The future results are based on the ensemble of eight GCMs. "D" and "W" indicate regions with drying and wetting trends, respectively.

Reference:
Caloiero T, Veltri S, Caloiero P, Frustaci F. 2018. Drought analysis in Europe and in the Mediterranean Basin using the standardized precipitation index. Water 10(8):1043

Hoerling, M., Eischeid, J., Perlwitz, J., Quan, X., Zhang, T., Pegion, P. 2012. On the increased frequency of Mediterranean drought. Journal of climate, 25(6), 2146-2161.

Peña-Angulo, D., Vicente-Serrano, S. M., Domínguez-Castro, F., Murphy, C., Reig, F., Tramblay, Y., Trigo, R. M., Luna, M. Y., Turco, M., Noguera, I., Aznárez-Balta, M., García-Herrera, R., Tomas-Burguera, M., and El Kenawy, A.: Long-term precipitation in Southwestern Europe reveals no clear trend attributable to anthropogenic forcing, Environ. Res. Lett., 2020. 15, 094070, https://doi.org/10.1088/1748-9326/ab9c4f.

Vicente-Serrano, S.M., Domínguez-Castro, F., Murphy, C., Hannaford, J., Reig, F., Peña-Angulo, D., Tramblay, Y., Trigo, R.M., Mac Donald, N., Luna, M.Y., Mc Carthy, M., Van der Schrier, G., Turco, M., Camuffo, D., Noguera, I., García-Herrera, R., Becherini, F., Della Valle, A., Tomas-Burguera, M., El Kenawy, A. 2021. Long-term variability and trends in meteorological droughts in Western Europe (1851–2018). Int. J. Climatol., 41, pp. E690-E717

Wagner, B., Vogel, H., Francke, A., Friedrich, T., Donders, T., Lacey, J. H., et al. 2019. Mediterranean winter rainfall in phase with African monsoons during the past 1.36 million years. Nature, 573(7773), 256–260. https://doi.org/10.1038/s41586-019-1529-0

(20) Lines 233-235: Do you mean that your results for these regions disagree with the previous studies? If yes, you could clarify a bit and discuss potential reasons for the disagreement. Also, you might want to replace "alternatively" with "on the contrary".

Response: We regret the misinterpretation. Our results are consistent with the previous studies and we have revised this statement from "wet to dry" to "dry to wet" in the new version. We also replaced "alternatively" with "on the contrary". The revised sentence is as follows (Lines 310-312):

On the contrary, some regions, such as the Amazon River basin, south Africa and eastern Australia, presenting wetting trends, are considered to experience a climatic shift from dry to the wet period (Chen et al., 2010; Gaughan and Waylen, 2012).

(21) Line 236: In SSP126 scenario, S. Europe also has a strong wetting trend.

Response: We have corrected this statement and elaborated more about the future

conditions over southwestern Europe in the revised version as follows (Lines 389-391):

Specifically, all three scenarios confirm the significant (p<0.05) wetting trends in North China, South Mongolia, central Asia, northern border of Canada, and South Europe, with the increase in the intensity and spread along with the enhancement of climate scenarios (Figures 1, 2, S14, and S15).

(22) Lines 236-245: This paragraph discusses only the SSP126 scenario, while the other two more probable scenarios remain uncommented. It would be nice to discuss the differences between each projection scenario and highlight the regions that all scenarios agree. Another striking difference appears in spatial clustering between the historical period and the model results in terms. It is evident that in the historical period there is stronger spatial homogeneity, while the models replicate this behavior only for SSP126 scenario. Any idea why this is happening?

Response: Thank you for the informative suggestion. We have added descriptions for the future projections under various scenarios and their similarities and differences in the revised version as follows (Lines 389-401):

Specifically, all three scenarios confirm the significant (p<0.05) wetting trends in North China, South Mongolia, central Asia, northern border of Canada, and South Europe, with the increase in the intensity and spread along with the enhancement of climate scenarios (Figures 1, 2, S14, and S15). Similarities are found in the drying trends in the majority of Russia, northern North America, and South Africa. The wetting trends are apparently caused by the increase in precipitation (Figure S16) (Milly et al., 2005; Seneviratne et al., 2006). The arid Arab region is also projected to become wetter because of the increase in precipitation and the decrease in evapotranspiration. On the contrary, the drying trends are mainly controlled by the rapidly intensifying evapotranspiration in a warming climate (Figure S17) (Allen et al., 2010; Vicente-Serrano et al., 2010), with the precipitation and runoff slightly increasing (Figures S16 and S18). The obvious drying trend around Canada's subarctic lakes is attributed to the high vulnerability to droughts when snow cover declines under increasing temperature (Bouchard et al., 2013). However, there exist scenario-variable divergences over the continents of South America, Australia, India, and the Mediterranean basin, which are generally caused by the various patterns in precipitation under different scenarios with the increasing evapotranspiration over there. The runoff also follows the patterns of precipitation but with comparably lesser magnitudes. The above-cited figures (Figures 1, 2, and S14-S18) are provided below for better comprehension.

Since the previous ensemble mean results of DATASET are mainly affected by three GRACE reconstructions (they are highly correlated with each other), the spatial distribution of trends shows spatial clustering. By presenting the individual results of each subset (Figure 1 below), we can only see this pattern in GRACE reconstruction from UTCSR, whereas the GHMs, LSMs, and GCMs show different spatial characteristics. We also mention this in the revised manuscript. Furthermore, we also

show the differences between the DDWW results in previous and present manuscripts in Table R1, suggesting the main conclusions are unaffected.

[Figure]

**Figure 1.** Global distribution of the classification in long-term trends in TWS-DSI during (a-f) the historical (1985-2014) and future (2071-2100) period under (g) SSP126, (h) SSP245, and (i) SSP585 scenarios. Note: The historical results are based on the (a) GRACE reconstruction, (b) WGHM, (c) VIC, (d) CLSM, (e) Noah, and (f) ensemble mean of eight GCMs, respectively. The future results are based on the ensemble of eight GCMs. "D" and "W" indicate regions with drying and wetting trends, respectively.

[Figure]

**Figure 2.** Global distribution of the classification in long-term trends in P-E-R during (a-f) the historical (1985-2014) and future (2071-2100) period under (g) SSP126, (h) SSP245, and (i) SSP585 scenarios. Note: The historical results are based on the (a) observation-based products (i.e., CRU P, GLEAM E, and GRUN R), (b) WGHM, (c) VIC, (d) CLSM, (e) Noah, and (f) ensemble mean of eight GCMs, respectively. The future results are based on the ensemble of eight GCMs. "D" and "W" indicate regions with drying and wetting trends, respectively.

[Figure]

**Figure S15.** Global distribution of the significant (p<0.05) long-term trends in P-E-R during (a-f) the historical (1985-2014) and future (2071-2100) period under (g) SSP126, (h) SSP245, and (i) SSP585 scenarios. Note: The historical results are based on the (a) observational products (i.e., CRU P-GLEAM E-GRUN R), (b) WGHM, (c) VIC, (d) CLSM, (e) Noah, and (f) ensemble mean of eight GCMs, respectively. The future results are based on the ensemble of eight GCMs.

[Figure]

**Figure S16.** Global distribution of the significant (p<0.05) long-term trends in P during (a-f) the historical (1985-2014) and future (2071-2100) period under (g) SSP126, (h) SSP245, and (i) SSP585 scenarios. Note: The historical results are based on the (a) CRU, (b) WGHM, (c) VIC, (d) CLSM, (e) Noah, and (f) ensemble mean of eight GCMs, respectively. The future results are based on the ensemble of eight GCMs. The VIC, CLSM, and Noah models are forced by the same precipitation dataset because they are from the GLDAS 2.0 family.

[Figure]

**Figure S17.** Global distribution of the significant (p<0.05) long-term trends in E during (a-f) the historical (1985-2014) and future (2071-2100) period under (g) SSP126, (h) SSP245, and (i) SSP585

scenarios. Note: The historical results are based on the (a) GLEAM E, (b) WGHM, (c) VIC, (d) CLSM, (e) Noah, and (f) ensemble mean of eight GCMs, respectively. The future results are based on the ensemble of eight GCMs.

[Figure]

**Figure S18.** Global distribution of the significant (p<0.05) long-term trends in R during (a-f) the historical (1985-2014) and future (2071-2100) period under (g) SSP126, (h) SSP245, and (i) SSP585 scenarios. Note: The historical results are based on the (a) GRUN, (b) WGHM, (c) VIC, (d) CLSM, (e) Noah, and (f) ensemble mean of eight GCMs, respectively. The future results are based on the ensemble of eight GCMs.

**Table R1.** Summary of the changes in the DDWW test results over global land during 1985-2014.

| Model/dataset | Previous results (ensemble mean of DATASET) | Updated results (individual datasets) [range] | Remark |
|---|---|---|---|
| DD | 16.7% | 6.47%-20.17% | From the perspective of TWSA, the DDWW is still challenged based on both the ensemble mean (previous version) and the individual datasets (current version) used in this study. |
| DW | 8.4% | 5.42%-16.13% | |
| WW | 11.4% | 4.54%-20.67% | |
| WD | 14.9% | 4.79%-19.3% | |
| TD | 2.1% | 0.95%3.88% | |
| TW | 1.8% | 0.73%-2.63% | |
| Non-significant | 45.1% | 17.2%-72.42% | |

(23) Line 245: You could consider rephrasing to "a pattern also considered".

Response: Rephrased as suggested.

(24) Lines 246-259: It is not very clear what the investigation of the changes over the SREX regions offers to the study.

Response: We have removed the case analysis based on the IPCC AR6 SREX regions in the revised manuscript while providing the processed data for the community to use in any regions (e.g., SREX, basins, etc.) of interest.

(25) Line 260: The legend is not very clear (some spaces between the numbers would help). Also, the scale order should be from higher to lower. Since the stippling marks are not very clear, could you please remove them and reproduce this map with only the statistically significant slopes in the supplementary material?

Response: We have revised the figures according to your suggestions in the revised version (Figures R6 and R7 above).

(26) Line 266: Same concerns here for the legend as the previous comment on Figure 1. The bar plot needs also revising as D should be above D (p<0.05). Additionally, I am not certain that the pie plots help the readers and are not discussed in the manuscript. You might want to consider removing them and adding them as a separate figure in the supplementary material.

Response: We have removed them for clarity in the revised manuscript as suggested.

(27) Lines 273-276: These lines would fit better to the Methods section.

Response: We have moved this paragraph to the Methods section as suggested.

(28) Line 274: Please elaborate about transitional regions.

Response: We have provided more information about the transitional regions in the revised manuscript as follows (Lines 234-237):
An approximate 7.9% of the land area is defined as the transitional region, referring to an intermediate between arid and humid climates. The transitional region generally lies in the shared boundaries of the humid and arid regions (e.g., western America, northern Canada, central Asia, western Africa, East Russia, and Australia).

(29) Line 290: You might want to remove "Under climate change" since you mention the SSP126 scenario.

Response: Removed.

(30) Lines 295-304: Again, I have similar concerns about SREX regions as the ones for lines 246-259. If you decide to keep them and justify the added value they offer in the analysis, please consider presenting these results in an individual paragraph.

Response: We have removed this content in the revised manuscript and have restrained our analysis on the global land only with additional regional analysis about the selected region of the Qinghai-Tibetan Plateau. Since we have provided the

processed data publicly available, further studies may focus on different regions of interest.

(31) Line 319: "In climate model projections", would be more appropriate than "Under climate change".

Response: We have changed it as suggested.

(32) Line 336: Please see the comment about Figure 2 (Line 266) regarding the pie charts.

Response: We have removed this figure according to the suggestions from Reviewer #1.

(33) Line 343: I would recommend using "Non-significant" instead of "Uncertain" here.

Response: We have used the term "Non-significant" throughout the revised manuscript.

(34) Line 348: It would be helpful to the readers to link the limitations with some suggestions for future research, especially for the first two paragraphs.

Response: As suggested, we have linked the inherent limitations of this study with future research (mainly related to the use of advanced bias correction methods, and including a larger number of GCM outputs as and when they are available) in the new version as follows (Lines 525-528 and 532-541, respectively):

…Overall, the models with completed TWS components are more suitable for assessing the TWSA changes at the global scale for future research, such as the continuously developing hyper-resolution global hydrological models (e.g., WGHM), which can help to avoid the uncertainty associated with the lack of key TWSA elements in most LSMs (e.g., surface water and groundwater) (Pokhrel et al., 2021).

…Advanced bias-correction methods (e.g., Lange, 2019 and Francois et al., 2020) might play critical roles in reducing such errors in meteorological variables for future hydrologic impact studies, especially when combined with the start-of-the-art GHMs and LSMs as mentioned above. The inclusion of more GCMs can also help to estimate the uncertainties in the meteorological inputs in climate change scenarios. Although it is challenging to explicitly attribute and quantify these uncertainties in the absence of a 'true' reference observation dataset, the ensemble averaging method has been used to integrate the multi-source TWSA data. Moreover, since the meaning and hence the results and interpretation of the 'dry' and 'wet' varies across disciplines, land or ocean, target variable(s), and the problem in question (Roth et al., 2021), future studies may focus on various spatial (e.g., local, regional, basin, zonal averages) and temporal (monthly, seasonal, annual) scales using our processed data with additional model outputs (e.g., more number of GCMs).

(35) Lines 379-385: Similarly to lines 190-208, bias-correction comes with certain limitations which need to be mentioned here.

Response: Since we have conducted independent comparisons between bias-corrected CMIP6 TWSA changes and observation-based water balance estimates (i.e., P-E-R), we think such limitations of the bias correction method using the GRACE data may not exist anymore.

(36) Line 382: A minor typo here "bias correction".

Response: We regret the typo. We have corrected it as suggested.

(37) Lines 403-407: It would be preferable to present the differences to the 0.05 threshold both in text and Figure S13.

Response: We have provided the difference between the 0.05 significance level to the 0.01 and 0.1 thresholds both in the text and supplementary file (same as Tables R1 and R2 below).

**Table R1.** Differences between the DDWW test results at 0.01 and 0.05 significance levels.

| Model/dataset | GRACE Reconstructions | WGHM | VIC | CLSM | Noah | GCM (historical period) | GCM (SSP126) | GCM (SSP245) | GCM (SSP585) |
|---|---|---|---|---|---|---|---|---|---|
| DD | -0.98% | -2.06% | -2.78% | -3.86% | -3.40% | -2.92% | -2.91% | -3.00% | -2.73% |
| DW | -0.84% | -2.36% | -3.19% | -2.24% | -2.45% | -2.68% | -2.46% | -2.40% | -2.74% |
| WW | -2.23% | -3.93% | -2.79% | -3.34% | -3.62% | -2.92% | -2.70% | -2.19% | -2.63% |
| WD | -1.81% | -2.43% | -4.04% | -3.99% | -3.31% | -3.94% | -3.79% | -4.64% | -4.00% |
| TD | -0.27% | -0.36% | -0.64% | -0.67% | -0.52% | -0.49% | -0.58% | -0.59% | -0.57% |
| TW | -0.30% | -0.52% | -0.41% | -0.52% | -0.58% | -0.54% | -0.45% | -0.39% | -0.54% |
| Non-significant | 6.42% | 11.65% | 13.84% | 14.61% | 13.89% | 13.48% | 12.88% | 13.21% | 13.20% |

**Table R2.** Differences between the DDWW test results at 0.1 and 0.05 significance levels.

| Model/dataset | GRACE Reconstruction | WGHM | VIC | CLSM | Noah | GCM (historical period) | GCM (SSP126) | GCM (SSP245) | GCM (SSP585) |
|---|---|---|---|---|---|---|---|---|---|
| DD | 0.52% | 1.46% | 1.97% | 2.14% | 2.23% | 1.82% | 1.85% | 1.89% | 1.60% |
| DW | 0.48% | 1.75% | 1.85% | 1.42% | 1.54% | 1.79% | 1.71% | 1.59% | 1.54% |

| | | | | | | | | | |
|---|---|---|---|---|---|---|---|---|---|
| WW | 1.13% | 3.12% | 2.19% | 2.04% | 2.24% | 2.16% | 2.07% | 1.74% | 1.72% |
| WD | 0.98% | 1.98% | 2.64% | 2.94% | 2.39% | 2.80% | 2.76% | 3.17% | 2.45% |
| TD | 0.20% | 0.45% | 0.44% | 0.73% | 0.35% | 0.43% | 0.46% | 0.36% | 0.43% |
| TW | 0.09% | 0.38% | 0.37% | 0.39% | 0.46% | 0.30% | 0.31% | 0.31% | 0.31% |
| Non-significant | -3.39% | -9.14% | -9.47% | -9.66% | -9.22% | -9.28% | -9.16% | -9.06% | -8.04% |

(38) Lines 421-424: A quite strong statement appears here. Are there any other studies that support it or is it derived only by the results of this study?

Response: Because we have presented the individual results of each subset of the DATASET and compared them with the corresponding metric P-E-R, we have weakened this statement in the new version as follows (Lines 567-568):
Given the inherent magnitude bias from various GCMs projections, the ensemble averaging method has the potential to provide alternative estimates over data-sparse areas globally like Africa and central Asia.

(39) Line 457: Another minor typo "is still challenged".

Response: Corrected as suggested.

(40) Line 463: It would be very beneficial to the community to share the data used in the manuscript figures, as well as the DATASET and bias-corrected CMIP6 members. This will have a positive impact on the study itself, as it will improve its reproducibility.

Response: We have made the data used in all the figures of the manuscript as well as the processed datasets and bias-corrected CMIP6 members publicly available via the Zenodo platform, which we will provide towards the later stages of the review.

**Reply to Reviewers' comments (Reviewer#3)**

**Legend**

Reviewers' comments

Authors' responses

Direct quotes from the revised manuscript

We thank the reviewer for his/her time in reading our manuscript and for detailed comments on our manuscript. Point-by-point replies to the comments or suggestions made can be found below. Overall, we have made the following major changes to the manuscript:

- Performed additional analysis using P-E-R and analyzed and compared the results with TWSA-DSI.
- Instead of showing only the ensemble mean of various model and observation-based results, we have now shown the results from individual datasets during the historical period (1985-2014).
- Added extensive discussion about the various mechanisms and governing processes for the observed patterns and the similarities and disparities from the previous studies.

**Reviewer #3:** This is my second review of the manuscript, which has been resubmitted after the previous discussion round.

The authors present a re-examination of the dry gets drier, and wet gets wetter paradigm over global land, based on terrestrial water storage estimates from different sources. They make use of GRACE reconstructions, global hydrological models, and land surface models, as well as CMIP6 models for the future perspective. They conclude that the DDWW paradigm is challenged both in the historical period but also in the future.

Overall, the authors took into account my points previously made and the manuscript considerably improved compared to the initial submission. In particular, the authors corrected the calculation of the percentage values by area-weighing the grid boxes and added more discussion on the uncertainties inherent in their analysis.

As such, I'm happy with the changes made. However, there are still some open methodological points that need to be addressed (see specific comments). Also, in parts the manuscript might need to be checked for grammar and wording by a native speaker.

Response: We thank the reviewer for recognizing the potential of the manuscript's new perspective, highly encouraging feedback, and his/her detailed suggestions for improvement. All the concerns raised have been addressed in the revised manuscript. We hope the modified text, along with the supplementary analyses and discussions, will put forward the results in a much more robust way.

**Specific comments:**

(1) Line 81: I suggest changing the title to "Data pre-processing"

Response: Changed as suggested.

(2) Line 84: Change to "(see Table 1 and next sections)"

Response: Thank you. We have changed it as suggested.

(3) Line 86: Change to "resampled to 1° x 1° resolution to compare against the average"

Response: Changed as suggested.

(4) Line 89: Change to "As for DATASET, the members of the CMIP6 ensemble have been resampled to …"

Response: We have merged this sentence with the previous line as follows (Lines 108-110):

The members of the CMIP6 ensemble and the all of the historical datasets have been resampled to 1°×1° scale using a bilinear interpolation approach for consistency and better comparison in the spatial domain.

(5) Line 106: "We have implemented an ensemble …" It's not clear in this sentence that you take the mean of the reconstructions derived from the three forcing datasets. Please rephrase.

Response: We have removed this statement since only the CSR reconstructions from Li et al. (2021) are used for the evaluation of the DDWW paradigm in the new version owing to high reliability and robustness (produced using a combination of three data-driven approaches and compared against various hydroclimatic indices and water storage component outputs by global hydrological models) of this product compared to the other ones (e.g., JPL and GSFC reconstructions from Humphrey and Gudmundsson (2019)) (Li et 2020; 2021)

References:

Humphrey, V., Gudmundsson, L., 2019. GRACE-REC: a reconstruction of climate-driven water storage changes over the last century. Earth Syst. Sci. Data 11, 1153–1170. https://doi.org/10.5194/essd-11-1153-2019

Li, F., Kusche, J., Chao, N., Wang, Z., Loecher, A., 2021. Long-Term (1979-Present) Total Water Storage Anomalies Over the Global Land Derived by Reconstructing GRACE Data. Geophys. Res. Lett. 48, e2021GL093492. https://doi.org/10.1029/2021GL093492

Li, F., Kusche, J., Rietbroek, R., Wang, Z., Forootan, E., Schulze, K., Lück, C. 2020. Comparison of Data-driven Techniques to Reconstruct (1992-2002) and Predict (2017-2018) GRACE-like Gridded Total Water Storage Changes using Climate Inputs. Water Resources Research, 56(5), e2019WR026551. https://doi.org/10.1029/2019wr026551.

(6) Line 107/108: Why only listing a subset of the variables used for the derivation of the CSR reconstruction and not all?

Response: We only listed a subset of input variables for CSR reconstruction since the other two reconstructions trained with JPL and GSFC are not used anymore, as we explained previously.

(7) Line 109: Change to "these three GRACE reconstructions"

Response: We changed it to "the GRACE reconstruction" since only one subset of GRACE reconstruction is used.

(8) Line 154: Change to "for which TWSA outputs are"

Response: Changed as suggested.

(9) Line 158: "and future periods (Krishnan and Bhaskaran, 2020)" Why citing this specific paper about wind speed in the Bay of Bengal? This does not appear to be the standard reference for CMIP6. Please use Eyring et al. 2016 instead.

Response: Thank you for the suggestion. We have updated the reference to Eyring et al. 2016.

(10) Line 162/163: "the sum of total soil moisture and snow water, which has been proven reliable to assess the TWS changes". This is already mentioned a few lines above, please merge and rephrase.

Response: As suggested, we have merged this statement with previous lines in the new version.

(11) Line 192/193: Change to "… (NRMSE) between the mean GRACE TWSA and the ensemble means of DATASET and CMIP6 data after bias correction during the period April 2002-December 2014, with the NRMSE calculated as the ratio of RMSE …"

Response: Thank you for the suggestion. We have changed it as suggested.

(12) Line 193: What is meant with "change range" of TWSA? The range of the TWSA values (i.e., max. minus min.) from DATASET and CMIP6 respectively? Please specify.

Response: Yes, it means the differences between the maximum and minimum TWSA. We have specified it in the revised version.

(13) Line 201: Change to "uncertainties in the CMIP6 simulations that remain even after undergoing the bias correction"

Response: Thank you for the suggestion. We have changed it as suggested.

(14) Line 210: Change to "The GRACE TWSA ranges from roughly −20 to 20 mm and shows …"

Response: Changed as suggested.

(15) Line 211: Change to "A similar temporal pattern is captured …"

Response: Thank you for the suggestion. We have changed it as suggested.

(16) Line 214/215: Change to "Moreover, the fluctuation range of DATASET is generally greater than the CMIP6 range before 2010. After 2010, DATASET tends to underestimate TWSA compared to CMIP6 and GRACE, and shows an increase in range."

Response: Since we have excluded the usage of the specific datasets (e.g., two other GRACE reconstructions from JPL and GSFC), the apparent underestimation of the datasets disappear now. So we have changed this sentence as follows (Lines 256-258): The GRACE TWSA ranges from roughly −20 to 20 mm and shows obvious seasonal characteristics with relatively higher uncertainty in the dry season than that in the wet season. A similar temporal pattern is captured by various models, with the change spread covering the variations of GRACE data.

(17) Line 217/218: Please rephrase, simplify, or merge with the following sentence. You look at long-term trends here.

Response: We have merged this sentence with the following text to focus on the long-term trends here.

(18) Line 219: "the historical period 1985-2014" Please clarify that the trend estimate for the historical period is based on DATASET (both in the text and in the figure captions). It would be interesting to see how the CMIP6 historical trends compare with the ones from DATASET.

Response: We have clarified the data source for the trend estimate during the historical and future periods in both text and figure captions. Moreover, we have compared the CMIP6 historical trends with the DATASET in Figures R1 and R2. The CMIP6 GCMs reasonably capture the decreasing trends in North America, Northeast Asia, North Africa, and South Australia together with increasing trends over central Africa, eastern Russia, and central North America, which are consistent with the DATASET. However, it also presents differently increasing TWS-DSI in central Asia and eastern Europe as well as depletion of South Africa. The differences might be arising from the bias in the atmospheric forcing and different TWSA components within these models. Based on this newly added comparison among CMIP6 historical trends and the ones from different models/products, we have appended the discussion in the revised manuscript.

[Figure]

**Figure R1** Global distribution of the significant (p<0.05) long-term trends in TWS-DSI during (a-f) the historical (1985-2014) and future (2071-2100) period under (g) SSP126, (h) SSP245, and (i) SSP585 scenarios. Note: The historical results are based on the (a) GRACE reconstruction, (b) WGHM, (c) VIC, (d) CLSM, (e) Noah, and (f) ensemble mean of eight GCMs, respectively. The future results are based on the ensemble of eight GCMs.

[Figure]

**Figure R2** Global distribution of the classification in long-term trends in TWS-DSI during (a-f) the historical (1985-2014) and future (2071-2100) period under (g) SSP126, (h) SSP245, and (i) SSP585 scenarios. Note: The historical results are based on the (a) GRACE reconstruction, (b) WGHM, (c) VIC, (d) CLSM, (e) Noah, and (f) ensemble mean of eight GCMs, respectively. The future results are based on the ensemble of eight GCMs. "D" and "W" indicate regions with drying and wetting trends, respectively.

(19) Line 237: Change to "become wetter because"

Response: Changed as suggested.

(20) Line 261, Figure 1: Please clarify in the caption what data source (i.e., DATASET or CMIP6) is used for the individual temporal subset.

Response: We have clarified the data source for the trends estimations during individual temporal subsets (Figures R1 and R2 above).

(21) Line 269, Figure 2: Change to "the bar plot displays the global percentage."

Response: We have removed this Figure according to the suggestions from Reviewers

**1 and #2.**

(22) Line 284: "Figure 3 illustrates the test results of DDWW paradigm at a 5% significance level (p=0.05)" Based on what test? Does the mentioned significance level relate to the test results for long-term trends? Please clarify how you derive these results in the methods section.

Response: The significance of the trend estimates is based on the modified Mann-Kendall test at a 5% significance level to avoid autocorrelation (Hamed and Rao, 1998). As rightly recognized by the reviewer, the mentioned significance here for DDWW evaluation indicates the test results for long-term trends, which has been clarified in the methods section as follows (Lines 217-223):

We convert the monthly TWS-DSI into annual means to calculate the long-term trends using the linear regression method. We examine the first-order autocorrelation of each TWSA dataset using the Durbin-Watson test (Durbin and Watson, 1950, 1951) (Figure S1). We find a total of 20% (GRACE reconstruction), 43% (WGHM), 41% (VIC), 23% (CLSM), 29% (Noah), and 20% (GCM) of the grid cells not presenting autocorrelation during 1985-2014, respectively. For the future period, the percentage is 25%, 26%, and 22% under the SSP126, SSP245, and SSP585 scenarios, respectively. In this case, the significance of the long-term trends is evaluated using the modified Mann-Kendall trend test at a 5% level to avoid autocorrelation (Hamed and Rao, 1998).

[Figure]

**Figure S1** Global assessment of the autocorrelation during the (a-f) historical (1985-2014) and future (2071-2100) period under (b) SSP126, (c) SSP245, and (d) SSP585 scenarios. Note: The historical results are based on the (a) GRACE reconstruction, (b) WGHM, (c) VIC, (d) CLSM, (e) Noah, and (f) ensemble mean of eight GCMs, respectively using the Durbin-Watson test. The future results are based on the ensemble of eight GCMs. Generally, the residuals are considered not auto-correlated when the Durbin-Watson test statistic has a value between 1.5 and 2.5. If the statistic is below 1 or above 3, then there is definitely autocorrelation among the residuals.

Reference:
Hamed, K. H., Rao, A. R. 1998. A modified Mann-Kendall trend test for autocorrelated data. Journal of hydrology, 204(1-4), 182-196.

(23) Line 332, Figure 3: Please clarify the data source (i.e., DATASET or CMIP6) used for the different temporal subsets. It would be interesting to see how the results

for DDWW based on the two data sources compare during the historical period. This could help to shed more light on the applicability of the CMIP6 ensemble for investigating the DDWW paradigm also in future periods.

Response: We have clarified the data source used for different temporal subsets in Figure R3. Moreover, the comparisons between the CMIP6 ensemble and different datasets during the historical period (1985-2014) are also performed. The CMIP6 historical results compare well with the multiple models. Specifically, the DD regions are mainly in North Africa, Northeast Asia, Arab region, and southwestern America. The WW area is generally located in the eastern Russia and coastal regions of West Africa. However, the CMIP6 data seems to fail to identify the WW pattern in the Spratly Islands of the South China Sea as well as Malaysia and Philippines. Moreover, the DD pattern is also slightly alleviated in central Asia. Overall, the CMIP6 ensembles show the potential to detect the large-scale changes in dryness/wetness globally, at least for the long-term trends by comparing with different data sources (i.e., GRACE reconstruction, GHMs, and LSMs). Based on this newly added comparison among CMIP6 historical trends and the ones from the historical datasets, we have appended the discussion in the revised manuscript.

[Figure]

**Figure R3** Global assessment of the DDWW paradigm during the (a) historical (1985-2014) and future (2071-2100) period under (b) SSP126, (c) SSP245, and (d) SSP585 scenarios. Note: The historical results are based on the (a) GRACE reconstruction, (b) WGHM, (c) VIC, (d) CLSM, (e) Noah, and (f) ensemble mean of eight GCMs, respectively. The future results are based on the ensemble of eight GCMs. DD indicates the dry gets drier; DW indicates the dry gets wetter; WW indicates the wet gets wetter; WD indicates the wet gets drier; TD indicates the transition gets drier; TW indicates the transition gets wetter.

(24) Line 340, Figure 4: "D" and "W" indicate regions with drying and wetting trends, respectively." I guess this does not belong to this figure caption?

Response: We have removed this Figure according to the suggestions from Reviewer #1 and #2.

(25) Line 364: "reported to show underestimation or overestimation" -> variable-specific biases?

Response: Thank you for the suggestion. We have changed it as suggested.

(26) Line 421/422: "Despite the magnitude bias from satellite products, simulations of LSMs and GHMs, and GCMs projections, …" Not sure what is meant here? Please rephrase.

Response: We have rephrased and simplified this statement given that we opt to present the DDWW evaluation results of each subset of datasets instead of only the ensemble mean.

(27) Line 453: "significance levels from 0.01 to 0.1" For the test on long-term trends?

Response: Yes, it indicates the test on long-term trends. We have revised this sentence in the new version for clarity.

---

## Author Response (AR2)

**Cover Letter**

Dear Editor,

We have further revised our manuscript according to the reviewer's detailed suggestions. We are pleased to resubmit the revised version of our manuscript titled "*Global evaluation of the dry gets drier and wet gets wetter paradigm from terrestrial water storage changes perspective*" (#ID: **hess-2022-190**).

All the comments are addressed in the new version of the manuscript to further improve the study. Please find below the attached point-by-point explanation of our correspondence for each suggestion by the reviewers. All the additional and changed parts of the text (except some minor language corrections) are marked in BLUE for easy review.

We sincerely hope you will find the revised version of the manuscript more comprehensive. All the authors have reviewed the manuscript and agree to the submission of the manuscript. We look forward to hearing from you.

Thank you for your time and efforts on our manuscript again.

Yours sincerely,

October 15th, 2022
Prof. Shenglian Guo
Corresponding author
State Key Laboratory of Water Resources and Hydropower Engineering Science,
Wuhan University, Wuhan 430072, P. R China
**E-mail:** slguo@whu.edu.cn

**Legend**

Reviewers' comments

Authors' responses

Direct quotes from the revised manuscript

We thank the reviewer for his/her time in reading our manuscript and detailed comments on our manuscript. Point-by-point replies to the comments made can be found below.

**Reviewer #1:** The authors perform a global examination for the dry gets dryer wet gets wetter paradigm from water storage change perspective using GRACE and various land surface/climate models. The method has limitations in glacier-covered regions, but it has the advantages of taking into account the effects of reservoir construction and water movement in the soil, etc. The authors have improved this study significantly. Most of my concerns are generally well resolved and explained. There are still some places that need to be modified.

Response: We thank the reviewer for his/her time in reviewing our manuscript with informative suggestions and recognizing the potential of the manuscript. Both implications and limitations of our study have been clearly claimed as suggested. Please find the subsequent minor modifications in the new version as follows.

1. Line 9: "...is still unexplored from the perspective of terrestrial water storage anomaly (TWSA)". Please add a "systematically" or "comprehensively".

Response: Added "comprehensively" as suggested.

2. Line 17-18: "...while the varying significance levels (0.01-0.1) have subtle influences on the evaluation results of the DDWW paradigm." It doesn't make sense to show this result

Response: As suggested, we have removed this sentence from the Abstract.

3. Line 72-73: "However, there is no study to examine the global variability and validity of DDWW paradigm in the past and future in terms of TWS changes ". This sentence is not very accurate.

Response: As suggested in *Comment #1*, we have added "comprehensively" for conveying the meaning better.

4. Line 91: An explanation on why a regional case study of QTP is needed here.

Response: We have clarified the reasons for choosing the QTP as a case study in the revised version as follows (Line 91-94):

One of the global hotspots with significant changes in hydroclimatological conditions (e.g., precipitation and air temperature) (Liu et al., 2006; Zhang et al., 2017), i.e., the Qinghai-Tibetan Plateau (QTP), is selected as a typical region for regional analysis because it experienced alarming TWS losses in recent decades and shows continuing declines under future scenarios (Meng et al., 2019; Li et al., 2022).

5. Line 284: "access" --> assess.

Response: Changed.

6. Line 307-310: "On the contrary...On the contrary". The logic of these sentences need have a revision.

Response: We have removed the repeating phrase "On the contrary" to make these sentences smoother.

7. Line 584-592: At the beginning of this paragraph the author should explain what the purpose of conducting the following experiment is. Some of the expressions should be simplified.

Response: We have added explanations for the reason to conduct the experiment and also refine this paragraph as follows (Lines 589-597):

This study performs a global examination for the dry gets dryer wet gets wetter paradigm from terrestrial water storage perspective in the past and future. The historical TWS-DSI monthly time series over global land during 1985-2014 is calculated from two GHMs (VIC and WGHM), two LSMs (Noah and CLSM), and one GRACE reconstruction. In addition, future projections of TWS-DSI from 2071 to 2100 under SSP126, SSP245, and SSP585 scenarios are derived from the average of eight selected CMIP6 GCMs after bias-correction using GRACE observations. Further, the DDWW paradigm has been evaluated with a significance level of 0.05 from the perspective of terrestrial water storage change. We also establish the metric P-E-R based on multiple observational products and from the same models as the TWS-DSI for comparison. The uncertainty sourced from different choices of models, methods, and confidence levels has been discussed systematically. The new findings are summarised as follows.

8. Line 607-616: These are experimental results or research procedures used for testing, and I don't think these are new findings and should not be taken as a conclusion. Line 617-620: I think these insights can be the third conclusion.

Response: We have removed these research procedures and reorganize the third

conclusion as follows:

(3) Sensitivity analysis on different choices of significance levels from 0.01 to 0.1 for the long-term trends indicates similar patterns, in which the maximum decrease (increase) in the DDWW-validated regions reaches –7.4% (4.47% historically under the 0.01 (0.1) level, respectively. Such consistency is also evidenced by the projected TWS-DSI in the future under various scenarios. Moreover, independent experiments based on the individual TWSA datasets suggest that the divergent data sources might lead to model-variable biases for both the DDWW-agreed and DDWW-opposed patterns. The use of distinctive GCMs also suggests slightly overrated (e.g., GFDL-ESM4) and underrated (e.g., CanESM5) percentages of such patterns in the future under multiple emission scenarios.

9. Line 620-622: Expressions about regional studies on the Qinghai-Tibet Plateau are not necessarily included in the conclusions.

Response: As suggested, we have shortened and reorganized results about the regional study of QTP in the conclusions as follows:

(2) A total of 11.01% (VIC) to 40.84% (GRACE reconstruction) of the global land area shows the DDWW paradigm valid, in which the drying and wetting area account for 6.47% (VIC)-20.17% (GRACE reconstruction) and 4.54% (VIC)-20.67% (GRACE reconstruction), respectively during the period 1985-2014. However, the area showing the opposite patterns, like "dry gets wetter" (DW) or "wet gets drier" (WD), account for the 10.21% (WGHM)-35.43% (GRACE reconstruction) of the global land, respectively. The proportion of areas supporting (opposing) the DDWW paradigm is 14.66% (16.76%), 14.26% (18.72%), and 17.08% (26.64%) under SSP126, SSP245, and SSP585 scenarios, respectively. Regional assessment for the QTP reveals the drying trends of the land mass primarily attributable to the sublimation/ablation of glaciers and ice caps, together with a continued tendency in future warming climates until the end of the 21st century.

10. Table S2: Are there references or standards for this classification (TWS-DSI)?

Response: We have added the references for the classification as Zhao et al. (2017).

Reference:
Zhao, M., Geruo, A., Velicogna, I., Kimball, J.S., 2017. Satellite Observations of Regional Drought Severity in the Continental United States Using GRACE-Based Terrestrial Water Storage Changes. J. Clim. 30, 6297–6308. https://doi.org/10.1175/JCLI-D-16-0458.1

11. The language still needs to be polished in the result analysis and the conclusion section.

Response: We have further polished our presentations in the new version.

**Reply to Reviewers' comments (Reviewer#2)**

**Legend**
Reviewers' comments
Authors' responses
Direct quotes from the revised manuscript

We thank the reviewer for his time in reading our manuscript. We hope the new changes could put the manuscript in a more robust way.

**Reviewer #2:** I would like to thank the authors for considering my comments and putting so much effort to address them convincingly. I have no further remarks.

Response: We thank the reviewer again for the enlightening suggestions and comments on our manuscript in previous reviews.

**Reply to Reviewers' comments (Reviewer#3)**

**Legend**

Reviewers' comments

Authors' responses

Direct quotes from the revised manuscript

We thank the reviewer for his/her time in reading our manuscript and for detailed comments on our manuscript. Point-by-point replies to the comments or suggestions made can be found below. Overall, we have made the following minor changes to the manuscript:

**Reviewer #3:** This is my third review of the manuscript.

The authors present a re-examination of the dry gets drier, and wet gets wetter paradigm over global land, based on terrestrial water storage estimates from different sources. They make use of GRACE reconstructions, global hydrological models, and land surface models, as well as CMIP6 models for the future perspective. They conclude that the DDWW paradigm is challenged both in the historical period but also in the future. In addition to the TWSA-based analysis, the revised version of the paper now also includes an analysis based on the water balance (P-E-R).

Overall, the authors considered my points previously made and the manuscript considerably improved.

As such, I'm happy with the changes made. A few remaining specific comments are listed in the following.

Response: We thank the reviewer for his/her informative and detailed suggestions and comments on our manuscript. Please find our changes for the remaining specific comments below.

**Specific comments:**

(1) Line 13: "with a 0.05 significance level" Specify that this significance level relates to the long-term trends. Similar on line 18.

Response: Changed as suggested.

(2) Line 217: "long-term trends using (Figure S1) the linear regression method". Reference to Figure S1 should be in the next sentence.

Response: Thank you. We have changed it as suggested.

(3) Line 237: "The DDWW paradigm is evaluated at a 5% significance level" Would be helpful to remind the reader that the significance level refers to the long-term trends.

Response: As suggested in *Comment #1 above,* we have specified the significance is associated to the trend estimates as follows (Line 238-239):

The DDWW paradigm is evaluated at a 5% significance level (trend estimates) in this study, combined with the standard AI-derived climate classifications.

(4) Line 255: "from different datasets" Here and at other locations in the text: why not explicitly stating what you look at? I.e., "GHMs and LSMs" instead of "different datasets".

Response: We have specified the datasets used for comparison here with the figure reference in the new version as follows (Lines 259-260):

A temporal comparison of global average TWSA derived from GHMs, LSMs, GRACE reconstruction, and CMIP6 and GRACE during 2002-2014 is shown in Figure S5.

(5) Line 257: "GRACE TWSA ranges … with relatively higher uncertainty in the dry season than that in the wet season." Figure S5 does not show the uncertainty for GRACE TWSA anymore, it's only displayed for the GCMs. Please adjust the figure or the text.

Response: The adjusted text is as below (Line 260-261):

The GRACE TWSA ranges from roughly −20 to 20 mm and shows obvious seasonal characteristics.

(6) Line 266: Figure S8 and S9 needs information in the caption on the content of the individual panels a-i.

Response: We have added panel information in the captions of Figures S8 and S9.

(7) Line 272: "present a strong correlation with the observational products before and after bias correction" What is the "observational products" here? The observed P-ET-R? Please provide more details in the figure caption and in general use a consistent naming of the different estimates throughout the text.

Response: Yes, it indicates the observed P-E-R here. We have revised the sentence and added more details in the caption of Figure S11. Moreover, we have added this information in the Table 1.

(8) Line 324: "magnitude of the changes in the water storage, i.e., TWSC, in a region are minimal compared to the actual TWSA trends" You could apply a temporal integration to convert TWSC into TWSA to omit this fact? Also, precipitation undercatch can influence the observation-based P-ET-R.

Response: Performing the temporal integration for TWSC can ideally provide useful information for the TWSA changes, however, the uncertainties sourced from various datasets like P, E, and R can make the true signals of long-term trends elusive. In the case, we did not calculate the TWSA from TWSC data and directly carry out the comparisons between them. So it is not surprising that there exist relatively huge differences between the TWSA and TWSC results. Lastly, we agree that biases in P (and E and R) can impact the evaluations of metric P-E-R, but the impacts are limited on our major outcomes since the metric is only used for comparison with our developed TWS-DSI results.

(9) Line 436: "As reported in Table R3, …" I guess this should be Table S3?

Response: Yes, and we have corrected it.

(10) Line 508: Figure 5 caption needs more information about the content of the panels a-f.

Response: We have added descriptions for the panels a-f in the caption of Figure 5.

(11) Line 607: Point 3 of your conclusions should be more explicit given the large differences between the TWSA-based analysis and the one from P-E-R. What are the main take-home messages here?

Response: According to *Comment #8* from **Reviewer #1**, we have removed these descriptions for research procedures from the conclusion in the new version.